# On the Convergence of Loss and Uncertainty-based Active Learning Algorithms

**Daniel Haimovich**
Meta, Central Applied Science
danielha@meta.com

**Dima Karamshuk**
Meta, Central Applied Science
karamshuk@meta.com

**Fridolin Linder**
Meta, Central Applied Science
flinder@meta.com

**Niek Tax**
Meta, Central Applied Science
niek@meta.com

**Milan Vojnović**
London School of Economics
m.vojnovic@lse.ac.uk

## Abstract

We investigate the convergence rates and data sample sizes required for training a machine learning model using a stochastic gradient descent (SGD) algorithm, where data points are sampled based on either their loss value or uncertainty value. These training methods are particularly relevant for active learning and data subset selection problems. For SGD with a constant step size update, we present convergence results for linear classifiers and linearly separable datasets using squared hinge loss and similar training loss functions. Additionally, we extend our analysis to more general classifiers and datasets, considering a wide range of loss-based sampling strategies and smooth convex training loss functions. We propose a novel algorithm called *Adaptive-Weight Sampling (AWS)* that utilizes SGD with an adaptive step size that achieves stochastic Polyak's step size in expectation. We establish convergence rate results for AWS for smooth convex training loss functions. Our numerical experiments demonstrate the efficiency of AWS on various datasets by using either exact or estimated loss values.

## 1 Introduction

In practice, when training machine learning models for prediction tasks (classification or regression), one often has access to an abundance of unlabeled data, while obtaining the corresponding labels may entail high costs. This may especially be the case in fields like computer vision, natural language processing, and speech recognition. Active learning algorithms are designed to efficiently learn a prediction model by employing a label acquisition, with the goal of minimizing the number of labels used to train an accurate prediction model.

Various label acquisition strategies have been proposed, each aiming to select informative points for the underlying model training task; including query-by-committee [Seung et al., 1992], expected model change [Settles et al., 2007], expected error reduction [Roy and McCallum, 2001], expected variance reduction [Wang et al., 2016], and mutual information maximization [Kirsch et al., 2019, Kirsch and Gal, 2022].

A common label acquisition strategy involves estimating uncertainty, which can be viewed as self-disagreement about predictions made by a given model. Algorithms using an uncertainty

38th Conference on Neural Information Processing Systems (NeurIPS 2024).

acquisition strategy are referred to as *uncertainty-based* active learning algorithms. Different variants of uncertainty strategies include margin of confidence, least confidence, and entropy-based sampling [Nguyen et al., 2022]. Recently, a *loss-based* active learning approach gained attention in research Yoo and Kweon [2019], Lahlou et al. [2022], Nguyen et al. [2021], Luo et al. [2021], and is now applied at scale in industry, such as for training integrity violation classifiers at Meta. This method involves selecting points for which there is a disagreement between the predicted label and the true label, as measured by a loss function. Since the true loss of a data point is unknown prior to the acquisition of the label, in practice, it is estimated using supervised learning. Loss-based sampling aligns with the spirit of the perceptron algorithm [Rosenblatt, 1958], which updates the model only for falsely-classified points.

Convergence guarantees for some uncertainty-based active learning algorithms have recently been established, such as for margin of confidence sampling [Raj and Bach, 2022]. By contrast, there are only limited results on the convergence properties of loss-based active learning algorithms, as these only recently been started to be studied, e.g., Liu and Li [2023].

The primary focus of this paper is to establish convergence guarantees for stochastic gradient descent (SGD) algorithms where points are sampled based on their loss. Our work provides new results on conditions that ensure certain convergence rates and bounds on the expected sample size, accommodating various data sampling strategies. Our theoretical results are under assumption that the active learner has access to an oracle that provides unbiased estimate of the conditional expected loss for a point, given the feature vector of the point and the current prediction model. In practice, the loss cannot be evaluated at acquisition time since labels are yet unknown. Instead, a separate prediction model is used for loss estimation. In our experiments, we assess the impact of the bias and noise in such a loss estimator. Our convergence rate analysis accommodates also uncertainty-based data selection, for which we provide new results.

Uncertainty and loss-based acquisition strategies are also of interest for the data subset selection problem, often referred to as core-set selection or data pruning. This problem involves finding a small subset of training data such that the predictive performance of a classifier trained on it is close to that of a classifier trained on the full training data. Recent studies have explored this problem in the context of training neural networks, as seen in works like Toneva et al. [2019], Coleman et al. [2020], Paul et al. [2021], Sorscher et al. [2022], Mindermann et al. [2022]. In such scenarios, the oracle can evaluate an underlying loss function exactly, avoiding the need for using a loss estimator.

There is a large body of work on convergence of SGD algorithms, e.g. see Bubeck [2015] and Nesterov [2018]. These results are established for SGD algorithms under either constant, diminishing or adaptive step sizes. Recently, Loizou et al. [2021], studied SGD with the stochastic Polyak's step size, depending on the ratio of the loss and the squared gradient of the loss of a point. Our work proposes an adaptive-window sampling algorithm and provides its convergence analysis, with the algorithm defined as SGD with a sampling of points and an adaptive step size update that conform to the stochastic Polyak's step size in expectation. This is unlike to the adaptive step size SGD algorithm by Loizou et al. [2021] which does not use sampling.

## 1.1 Summary of our Contributions

Our contributions can be summarizes as given in the following points:

• For SGD with a constant step size, we present conditions under which a non-asymptotic convergence rate of order $O(1/n)$ holds, where $n$ represents the number of iterations of the algorithm, i.e., the number of unlabeled points presented to the algorithm. These conditions enable us to establish convergence rate results for loss-based sampling in the case of linear classifiers and linearly separable datasets, with the loss function taking on various forms such as the squared hinge loss function, generalized hinge loss function, or satisfying other specified conditions. Our results provide bounds for both expected loss and the number of sampled points, encompassing different loss-based strategies. These results are established by using a convergence rate lemma that may be of independent interest.

• For SGD with a constant step size, we provide new convergence rate results for more general classifiers and datasets, with sampling of points according to an increasing function $\pi$ of the conditional expected loss of a point. In this case, we present conditions for smooth convex training loss functions under which a non-asymptotic convergence rate of order $O(\Pi^{-1}(1/\sqrt{n}))$ holds, where $\Pi$ is the primitive function of $\pi$. These results are established by leveraging the fact that the algorithm behaves

akin to a SGD algorithm with an underlying objective function, as referred to as an *equivalent loss* in Liu and Li [2023], allowing us to apply known convergence rate results for SGD algorithms.

• We propose *Adaptive-Weight Sampling (AWS)*, a novel learning algorithm that combines a sampling-based acquisition strategy with an adaptive step-size SGD update, achieving the stochastic Polyak's step size update in expectation, which can be used with any differentiable loss function. We establish a condition under which a non-asymptotic convergence rate of order $O(1/n)$ holds for AWS with smooth convex loss functions. We present uncertainty and loss-based strategies that satisfy this condition for binary classification, as well as an uncertainty strategy for multi-class classification.

• We present numerical results that demonstrate the efficiency of AWS on various datasets.

## 1.2 Related Work

The early proposal of the query-by-committee (QBC) algorithm by [Seung et al., 1992] demonstrated the benefits of active learning, an analysis of which was conducted under the selective sampling model by Freund et al. [1997] and Gilad-bachrach et al. [2005]. Dasgupta et al. [2009] showed that the performance of QBC can be efficiently achieved by a modified perceptron algorithm with adaptive filtering. The efficient and label-optimal learning of halfspaces was studied by Yan and Zhang [2017] and, subsequently, by Shen [2021]. Online active learning algorithms, studied under the name of selective sampling, include works by [Cesa-Bianchi et al., 2006, 2009, Dekel et al., 2012, Orabona and Cesa-Bianchi, 2011, Cavallanti et al., 2011, Agarwal, 2013]. For a survey, refer to Settles [2012].

Uncertainty sampling has been utilized for classification tasks since as early as [Lewis and Gale, 1994], and subsequently in many other works, such as [Schohn and Cohn, 2000, Zhu et al., 2010, Yang et al., 2015, Yang and Loog, 2016, Lughofer and Pratama, 2018]. Mussmann and Liang [2018] demonstrated that threshold-based uncertainty sampling on a convex loss can be interpreted as performing a pre-conditioned stochastic gradient step on the population zero-one loss. However, none of these works have provided theoretical convergence guarantees.

The convergence of margin of confidence sampling was recently studied by Raj and Bach [2022], who demonstrated linear convergence for linear classifiers and linearly separable datasets, specifically for the hinge loss function, for a family of selection probability functions and an algorithm that performs a SGD update with respect to the squared hinge loss function. However, our results for linear classifiers and linearly separable datasets differ, as our focus lies on loss-based sampling strategies and providing bounds on the convergence rate of a loss function and the expected number of sampled points. These results are established using a convergence rate lemma, which may be of independent interest. It is noteworthy that the convergence rate for uncertainty-based sampling, as in Theorem 3.1 of Raj and Bach [2022], can be derived by checking the conditions of the convergence rate lemma.

A loss-based active learning algorithm was proposed by Yoo and Kweon [2019], comprising a loss prediction module and a target prediction model. The algorithm uses the loss prediction module to compute a loss estimate and prioritizes sampling points with a high estimated loss under the current prediction model. Lahlou et al. [2022] generalize this idea within a framework for uncertainty prediction. However, neither Yoo and Kweon [2019] nor Lahlou et al. [2022] provided theoretical guarantees for convergence rates. Recent analysis of convergence for loss and uncertainty-based active learning strategies has been presented by Liu and Li [2023]. Specifically, they introduced the concept of an equivalent loss, demonstrating that a gradient descent algorithm employing point sampling can be viewed as a SGD algorithm optimizing an equivalent loss function. While they focused on specific cases like sampling proportional to conditional expected loss, our results allow for sampling based on any continuous increasing function of expected conditional loss, and provide explicit convergence rate bounds in terms of the underlying sampling probability function.

In addition, Loizou et al. [2021] introduced a SGD algorithm featuring an adaptive stochastic Polyak's step size, which has theoretical convergence guarantees under various assumptions. This algorithm showcased robust performance in comparison to state-of-the-art optimization methods, especially when training over-parametrized models. Our work proposes a novel sampling method that employs stochastic Polyak's step size in expectation, offering a convergence rate guarantee for smooth convex loss functions, contingent on a condition related to the sampling probability function. Notably, we demonstrate the fulfillment of this condition for logistic regression and binary cross-entropy loss functions, encompassing both a loss-based strategy involving proportional sampling to absolute error

loss and an uncertainty sampling strategy. Furthermore, we extend this condition to hold for an uncertainty sampling strategy designed for multi-class classification.

## 2 Problem Statement

We consider the setting of streaming algorithms where a machine learning model parameter $\theta_t$ is updated sequentially, upon encountering each data point, with $(x_1, y_1), \ldots, (x_n, y_n) \in \mathcal{X} \times \mathcal{Y}$ denoting the sequence of data points with the corresponding labels, assumed to be independent and identically distributed with distribution $\mathcal{D}$. Specifically, we consider the class of projected SGD algorithms defined as: given an initial value $\theta_1 \in \Theta$,

$$\theta_{t+1} = \mathcal{P}_{\Theta_0} \left( \theta_t - z_t \nabla_\theta \ell(x_t, y_t, \theta_t) \right), \text{ for } t \geq 1 \tag{1}$$

where $\ell : \mathcal{X} \times \mathcal{Y} \times \Theta \to \mathbb{R}$ is a training loss function, $z_t$ is a stochastic step size with mean $\zeta(x_t, y_t, \theta_t)$ for some function $\zeta : \mathcal{X} \times \mathcal{Y} \times \Theta \mapsto \mathbb{R}_+$, $\Theta_0 \subseteq \Theta$, and $\mathcal{P}_{\Theta_0}$ is the projection function, i.e., $\mathcal{P}_{\Theta_0}(u) = \arg\min_{v \in \Theta_0} ||u - v||$. Unless specified otherwise, we consider the case $\Theta_0 = \Theta$, which requires no projection. For binary classification tasks, we assume $\mathcal{Y} = \{-1, 1\}$. For every $t > 0$, we define $\bar{\theta}_t = (1/t) \sum_{s=1}^t \theta_s$.

By defining the distribution of the stochastic step size $z_t$ in Equation (1) appropriately, we can accommodate different active learning and data subset selection algorithms. In the context of active learning algorithms, at each step $t$, the algorithm observes the value of $x_t$ and decides whether or not to observe the value of the label $y_t$. The value of $z_t$ determine whether or not we observe the label $y_t$. Deciding not to observe the value of the label $y_t$ implies the step size $z_t$ of value zero (not updating the machine learning model).

For the choice of the stochastic step size, we consider two cases: (a) *Constant-Weight Sampling*: a Bernoulli sampling with a constant step size, and (b) *Adaptive-Weight Sampling*: a sampling that achieves stochastic Polyak's step size in expectation. For case (a), $z_t$ is the product of a constant step size $\gamma$ and a Bernoulli random variable with mean $\pi(x_t, y_t, \theta_t)$. For case (b), $\zeta(x, y, \theta)$ is the "stochastic" Polyak's step size, and $z_t$ is equal to $\zeta(x_t, y_t, \theta_t)/\pi(x_t, y_t, \theta_t)$ with probability $\pi(x_t, y_t, \theta_t)$ and is equal to 0 otherwise. Note that using the notation $\pi(x, y, \theta)$ allows for the case when the sampling probability does not depend on the value of the label $y$.

For a *loss-based sampling*, $\pi$ is an increasing function of some loss function $\ell^\star$, which does not necessarily correspond to the training loss function $\ell$. Specifically, for a binary classifier with $p(x, y, \theta)$ denoting the expected prediction label, *sampling proportional to the absolute error loss* is defined as $\pi(\ell^*) = \omega \ell^*$ where $\ell^*(x, y, \theta) = |y - p(x, y, \theta)|$ and $\omega \in (0, 1/2)$. For an *uncertainty-based sampling*, $\pi$ is a function of some quantity reflecting the uncertainty of the prediction model.

Our focus is on finding convergence conditions for algorithm (1) and convergence rates under these conditions, as well as bounds on the expected number of points sampled by the algorithm.

**Additional Assumptions and Notation**  For binary classification, we say that data is *separable* if, for every point $(x, y) \in \mathcal{X} \times \mathcal{Y}$, either $y = 1$ with probability 1 or $y = -1$ with probability 1. The data is *linearly separable* if there exists $\theta^* \in \Theta$ such that $y = \text{sgn}(x^\top \theta^*)$ for every $x \in \mathcal{X}$. Linearly separable data has a $\rho^*$-*margin* if $|x^\top \theta^*| \geq \rho^*$ for every $x \in \mathcal{X}$, for some $\theta^* \in \Theta$.

Some of our results are for *linear classifiers*, where the predicted label of a point $x$ is a function of $x^\top \theta$. For example, a model with a predicted label $\text{sgn}(x^\top \theta)$ is a linear classifier. For logistic regression, the predicted label is 1 with probability $\sigma(x^\top \theta)$ and $-1$ otherwise, where $\sigma$ is the logistic function defined as $\sigma(z) = 1/(1 + e^{-z})$. For binary classification, we model the prediction probability of the positive label as $\sigma(x^\top \theta)$, where $\sigma : \mathbb{R}_+ \to [0, 1]$ is an increasing function, and $\sigma(-u) + \sigma(u) = 1$ for all $u \in \mathbb{R}$. The absolute error loss takes value $1 - \sigma(x^\top \theta)$ if $y = 1$ or value $\sigma(x^\top \theta)$ if $y = -1$, which corresponds to $1 - \sigma(yx^\top \theta)$. The binary cross-entropy loss for a point $(x, y)$ under model parameter $\theta$ can be written as $\ell(x, y, \theta) = -\log(\sigma(yx^\top \theta))$. Hence, absolute error loss-based sampling corresponds to the sampling probability function $\pi(\ell) = 1 - e^{-\ell}$.

For any given $(x, y) \in \mathcal{X} \times \mathcal{Y}$, the loss function $\ell(x, y, \theta)$ is considered *smooth* on $\Theta' \subseteq \Theta$ if it has a Lipschitz continuous gradient on $\Theta'$, i.e., there exists $L_{x,y}$ such that $||\nabla_\theta \ell(x, y, \theta_1) - \nabla_\theta \ell(x, y, \theta_2)|| \leq L_{x,y} ||\theta_1 - \theta_2||$ for all $\theta_1, \theta_2 \in \Theta'$. For any distribution $q$ over $\mathcal{X} \times \mathcal{Y}$, $\mathbb{E}_{(x,y) \sim q}[\ell(x, y, \theta)]$ is $\mathbb{E}_{(x,y) \sim q}[L_{x,y}]$-smooth.

# 3 Convergence Rate Guarantees

In this section, we present conditions on the stochastic step size of algorithm (1) under which we can bound the total expected loss and the expected number of samples. For the Constant-Weight Sampling, we provide conditions that allow us to derive bounds for linear classifiers and linearly separable datasets and more general cases. For Adaptive-Weight Sampling, we offer a condition that allows us to establish convergence bounds for both loss and uncertainty-based sampling.

## 3.1 Constant-Weight Sampling

**Linear Classifiers and Linearly Separable Datasets** We focus on binary classification and briefly discuss extension to multi-class classification. We consider the linear classifier with the predicted label $\text{sgn}(x^\top \theta)$. With a slight abuse of notation, let $\ell(x, y, \theta) \equiv \ell(u)$ and $\pi(x, y, \theta) \equiv \pi(u)$ where $u = yx^\top \theta$. We assume that the domain $\mathcal{X}$ is bounded, i.e., there exists $R$ such that $||x|| \leq R$ for all $x \in \mathcal{X}$, $||\theta_1 - \theta^*|| \leq S$ for some $S \geq 0$, and that the data is $\rho^*$-margin linearly separable.

We present convergence rate results for the training loss function corresponding to the squared hinge loss function, i.e. $\ell(u) = (1/2) \max\{1 - u, 0\}^2$. Our additional results also cover other cases, including a class of smooth convex loss functions and a generalized smooth hinge loss function, which are presented in the Appendix.

**Theorem 3.1.** *Assume that $\rho^* > 1$, the loss function is the squared hinge loss function, and the sampling probability function $\pi$ is such that for all $u \leq 1$, $\pi(u) \leq \beta/2$ and*

$$\pi(u) \geq \pi^*(\ell(u)) := \frac{\beta}{2}\left(1 - \frac{1}{1 + \mu\sqrt{\ell(u)}}\right) \tag{2}$$

*for some constants $0 < \beta \leq 2$ and $\mu \geq \sqrt{2}/(\rho^* - 1)$. Then, for any initial value $\theta_1$ such that $||\theta_1 - \theta^*|| \leq S$ and $\{\theta_t\}_{t>1}$ according to algorithm (1) with $\gamma = 1/R^2$,*

$$\mathbb{E}\left[\ell(yx^\top \bar{\theta}_n)\right] \leq \mathbb{E}\left[\frac{1}{n}\sum_{t=1}^{n} \ell(y_t x_t^\top \theta_t)\right] \leq \frac{R^2 S^2}{\beta}\frac{1}{n},$$

*where $(x, y)$ is an independent sample of a labeled data point from $\mathcal{D}$.*

*Moreover, if the sampling is according to $\pi^*$, then the expected number of sampled points satisfies*

$$\mathbb{E}\left[\sum_{t=1}^{n} \pi^*(\ell(y_t x_t^\top \theta_t))\right] \leq \min\left\{\frac{1}{2}RS\mu\sqrt{\beta}\sqrt{n}, \frac{1}{2}\beta n\right\}.$$

Condition (2) requires that the sampling probability function $\pi$ is lower bounded by an increasing, concave function $\pi^*$ of the loss value. This fact, along with the expected loss bound, implies the asserted bound for the expected number of samples. The expected number of samples is $O(\sqrt{n})$ concerning the number of iterations and is $O(1/(\rho^* - 1))$ concerning the margin $\rho^* - 1$.

Theorem 3.1, and our other results for linear classifiers and linearly separable datasets, are established using a convergence rate lemma, which is presented in Appendix A.2, along with its proof. This lemma generalizes the conditions used to establish the convergence rate for an uncertainty-based sampling algorithm by Raj and Bach [2022], with the sampling probability function $\pi(u) = 1/(1 + \mu|u|)$, for some constant $\mu > 0$. It can be readily shown that Theorem 3.1 in Raj and Bach [2022] follows from our convergence rate lemma with the training loss function corresponding to the squared hinge loss function and the evaluation loss function (used for convergence rate guarantee) corresponding to the hinge loss function. Further details on the convergence rate lemma are discussed in Appendix A.2.1.

The convergence rate conditions for multi-class classification with the set of classes $\mathcal{Y}$ are the same as for binary classification, with $u(x, y, \theta) := x^\top \theta_y - \max_{y' \in \mathcal{Y}\setminus\{y\}} x^\top \theta_{y'}$, except for an additional factor of 2 in one of the conditions (see Lemma A.10 in the Appendix). Hence, all the observations remain valid for the multi-class classification case.

Table 1: Examples of sampling probability functions.

| $\pi(x)$ | $\Pi(x)$ | $\Pi^{-1}(x)$ |
|---|---|---|
| $1 - e^{-x}$ | $x + e^{-x} - 1$ | $\approx \sqrt{2x}$ for small $x$ |
| $\min\{x, 1\}$ | $\begin{cases} \frac{1}{2}x^2 & x \leq 1 \\ x - \frac{1}{2} & x \geq 1 \end{cases}$ | $\begin{cases} \sqrt{2x} & x \leq 1/2 \\ x + \frac{1}{2} & x \geq 1/2 \end{cases}$ |
| $\min\{(x/b)^a, 1\}, a > 0, b > 0$ | $\begin{cases} \frac{1}{b^a(1+a)}x^{1+a} & x \leq a \\ x - \frac{a}{1+a} & x \geq a \end{cases}$ | $\begin{cases} b^{\frac{a}{1+a}}(1+a)^{\frac{1}{1+a}}x^{\frac{1}{1+a}} & x \leq \frac{b}{1+a} \\ x + \frac{a}{1+a}b & x \geq \frac{b}{1+a} \end{cases}$ |
| $1 - \frac{1}{1+\mu x}$ | $x - \frac{1}{\mu}\log(1 + \mu x)$ | $\approx \sqrt{(2/\mu)x}$ for small $x$ |
| $1 - \frac{1}{1+\mu\sqrt{x}}$ | $x - \frac{2}{\mu}\sqrt{x} + \frac{2}{\mu^2}\log(1 + \mu\sqrt{x})$ | $\approx (((3/2)/\mu)x)^{2/3}$ for small $x$ |

**More General Classifiers and Datasets**  We consider algorithm (1) where $z_t$ is product of a fixed step size $\gamma$ and a Bernoulli random variable $\zeta_t$ with mean $\pi(x, y, \theta)$. Let $g_t = \zeta_t \nabla_\theta \ell(x_t, y_t, \theta_t)$, which is random vector because $\zeta_t$ is a random variable and $(x_t, y_t)$ is a sampled point. Following Liu and Li [2023], we note that the algorithm (1) is an SGD algorithm with respect to an objective function $\tilde{\ell}$ with gradient

$$\nabla_\theta \tilde{\ell}(\theta) = \mathbb{E}[\pi(x, y, \theta)\nabla_\theta \ell(x, y, \theta)] \tag{3}$$

where the expectation is with respect to $x$ and $y$. This observation allows us to derive convergence rate results by deploying convergence rate results that are known to hold for SGD under various assumptions on function $\tilde{\ell}$, variance of stochastic gradient vector and step size. A function $\tilde{\ell}$ satisfying condition (3) is referred to as an *equivalent loss* in Liu and Li [2023].

Assume that the sampling probability $\pi$ is an increasing function of the conditional expected loss $\ell(x, \theta) = \mathbb{E}_y[\ell(x, y, \theta) \mid x]$. With a slight abuse of notation, we denote this probability as $\pi(\ell(x, \theta))$ where $\pi : \mathbb{R}_+ \to [0, 1]$ is an increasing and continuous function. Let $\Pi$ be the primitive of $\pi$, i.e. $\Pi' = \pi$. We then have

$$\tilde{\ell}(\theta) = \mathbb{E}[\Pi(\ell(x, \theta))]. \tag{4}$$

If $\ell(x, y, \theta)$ is a convex function, for every $(x, y) \in \mathcal{X} \times \mathcal{Y}$, then $\tilde{\ell}$ is a convex function.

This framework for establishing convergence rates allows us to accommodate different sampling strategies and loss functions. The next lemma allows us to derive convergence rate results for expected loss with respect to loss function $\ell$ by applying convergence rate results for expected loss with respect to loss function $\tilde{\ell}$ (which, recall, is the equivalent loss function).

**Lemma 3.2.** *Assume that for algorithm (1) with loss-based sampling according to $\pi$, for some functions $f_1, \ldots, f_m$, we have*

$$\mathbf{E}\left[\frac{1}{n}\sum_{t=1}^{n}\tilde{\ell}(\theta_t)\right] \leq \inf_\theta \tilde{\ell}(\theta) + \sum_{i=1}^{m}f_i(n). \tag{5}$$

*Then, it holds:*

$$\mathbf{E}\left[\frac{1}{n}\sum_{t=1}^{n}\ell(\theta_t)\right] \leq \inf_\theta \Pi^{-1}(\tilde{\ell}(\theta)) + \sum_{i=1}^{m}\Pi^{-1}(f_i(n)).$$

We apply Lemma 3.2 to obtain the following result.

**Theorem 3.3.** *Assume that $\ell$ is a convex function, $\tilde{\ell}$ is $L$-smooth, $\Theta_0$ is a convex set, $S = \sup_{\theta \in \Theta_0} \|\theta - \theta_1\|$, and $\mathbb{E}[\pi(\ell(x, \theta))\|\nabla_\theta \ell(x, y, \theta)\|^2] - \|\nabla_\theta \tilde{\ell}(\theta)\|^2 \leq \sigma_\pi^2$. Then, for algorithm (1) with $\gamma = 1/(L + (\sigma/R)\sqrt{n/2})$,*

$$\mathbb{E}[\ell(\bar{\theta}_n)] \leq \mathbb{E}\left[\frac{1}{n}\sum_{t=1}^{n}\ell(\theta_t)\right] \leq \inf_\theta \Pi^{-1}(\tilde{\ell}(\theta)) + \Pi^{-1}\left(\frac{\sqrt{2}S\sigma_\pi}{\sqrt{n}}\right) + \Pi^{-1}\left(\frac{LS^2}{n}\right).$$

Note that the bound on the expected loss in Theorem 3.3 depends on $\pi$ through $\Pi^{-1}$ and $\sigma_\pi^2$. Specifically, we have a bound depending on $\pi$ only through $\Pi^{-1}$ by upper bounding $\sigma_\pi^2$ with $\sup_{\theta \in \Theta_0} \mathbb{E}[\|\nabla_\theta \ell(x, y, \theta)\|^2]$.

For convergence rates for large values of the number of iterations $n$, the bound in Theorem 3.2 crucially depends on how $\Pi^{-1}(x)$ behaves for small values of $x$. In Table 1, we show $\Pi$ and $\Pi^{-1}$ for several examples of sampling probability function $\pi$. For all examples in the table, $\Pi^{-1}(x)$ is sub-linear in $x$ for small $x$. For instance, for absolute error loss sampling under binary cross-entropy loss function, $\pi(x) = 1 - e^{-x}$, $\Pi^{-1}(x)$ is approximately $\sqrt{2x}$ for small $x$. For this case, we have the following corollary.

**Corollary 3.4.** *Under assumptions of Theorem 3.3, sampling probability $\pi(x) = 1 - e^{-x}$, and $n \geq \max \left\{ \left( \frac{16}{9} \right)^2 2S\sigma_\pi^2, \frac{16}{9}LS^2 \right\}$ it holds*

$$\mathbb{E}[\ell(\bar{\theta}_n)] \leq \mathbb{E}\left[ \frac{1}{n} \sum_{t=1}^n \ell(\theta_t) \right] \leq \inf_\theta \Pi^{-1}(\tilde{\ell}(\theta)) + 2^{5/4}\sqrt{S\sigma_\pi}\frac{1}{\sqrt[4]{n}} + 2\sqrt{L}S\frac{1}{\sqrt{n}}.$$

By using a bound on the expected total loss, we can bound the expected total number of sampled points under certain conditions as follows.

**Lemma 3.5.** *The following bounds hold:*

1. *Assume that $\pi$ is a concave function, then $\mathbb{E}\left[\sum_{t=1}^n \pi(\ell(x_t, \theta_t))\right] \leq \pi\left(\mathbb{E}\left[\frac{1}{n}\sum_{t=1}^n \ell(\theta_t)\right]\right) n.$*

2. *Assume that $\pi$ is $K$-Lipschitz or that for some $K > 0$, $\pi(\ell) \leq \min\{K\ell, 1\}$ for all $\ell \geq 0$, then $\mathbb{E}\left[\sum_{t=1}^n \pi(\ell(x_t, \theta_t))\right] \leq \min\left\{K\mathbb{E}\left[\sum_{t=1}^n \ell(\theta_t)\right], n\right\}.$*

We remark that $\pi$ is a concave function for all examples in Table 1 without any additional conditions, except for $\pi(\ell) = \min\{(\ell/b)^a, 1\}$ which is concave under assumption $0 < a \leq 1$. We remark also that for every example in Table 1 except the last one, $\pi(\ell) \leq \min\{K\ell, 1\}$ for some $K > 0$. Hence, for all examples in Table 1, we have a bound for the expected number of sampled points provided we have a bound for the expected loss.

## 3.2 Adaptive-Weight Sampling

In this section we propose the *Adaptive-Weight Sampling (AWS)* algorithm that combines Bernoulli sampling and an adaptive SGD update, and provide a convergence rate guarantee. The algorithm is defined by (1) with the stochastic step size $z_t$ being a binary random variable that takes value $\gamma_t := \zeta(x_t, y_t, \theta_t)/\pi(x_t, y_t, \theta_t)$ with probability $\pi(x_t, y_t, \theta_t)$ and takes value 0 otherwise, where $\pi$ is some sampling probability function. Here, $\zeta(x, y, \theta)$ is the expected SGD (1) step size, defined as

$$\zeta(x, y, \theta) = \beta \min \left\{ \frac{1}{\psi(x, y, \theta)}, \rho \right\}$$

whenever $||\nabla_\theta \ell(x, y, \theta)|| > 0$ and $\zeta(x, y, \theta) = 0$ otherwise, for constants $\beta, \rho > 0$, where

$$\psi(x, y, \theta) := \frac{||\nabla_\theta \ell(x, y, \theta)||^2}{\ell(x, y, \theta) - \inf_{\theta'} \ell(x, y, \theta')}.$$

The expected step size $\zeta(x, y, \theta)$ corresponds to the stochastic Polyak's step size used by a gradient descent algorithm proposed by [Loizou et al., 2021], which is accommodated as a special case when $\pi(x, y, \theta) = 1$ for all $x, y, \theta$. AWS introduces a sampling component and re-weighting of the update to ensure that the step size remains according to the stochastic Polyak's step size in expectation. For many loss functions, $\inf_{\theta'} \ell(x, y, \theta) = 0$, for every $x, y$. In these cases, $\psi(x, y, \theta) = ||\nabla_\theta \ell(x, y, \theta)||^2/\ell(x, y, \theta)$. For instance, for binary cross-entropy loss function, $\inf_{\theta'} \ell(x, y, \theta) = \inf_{\theta'}(-\log(\sigma(yx^\top\theta'))) = 0$, for all $x, y$.

We next show a convergence rate guarantee for AWS.

**Theorem 3.6.** *Assume that $\ell$ is a convex, $L$-smooth function, there exists $\Lambda^*$ such that $\mathbb{E}[\ell(x, y, \theta^*)] - \mathbb{E}[\inf_\theta \ell(x, y, \theta)] \leq \Lambda^*$, and the sampling probability function $\pi$ is such that, for some constant $c \in (0, 1)$, for all $x, y, \theta$ such that $||\nabla_\theta \ell(x, y, \theta)|| > 0$,*

$$\pi(x, y, \theta) \geq \frac{\beta}{2(1-c)} \min \left\{ \rho\psi(x, y, \theta), 1 \right\}. \tag{6}$$

*Then, we have*

$$\mathbb{E}\left[\frac{1}{n}\sum_{t=1}^{n}(\ell(x_t, y_t, \theta_t) - \ell(x_t, y_t, \theta_t^*))\right] \leq \frac{\rho\beta}{c\kappa}\Lambda^* + \frac{1}{2c\kappa}||\theta_1 - \theta^*||^2\frac{1}{n}$$

*where $\kappa = \beta\min\{1/(2L), \rho\}$ and $\theta_t^*$ is a minimizer of $\ell(x_t, y_t, \theta')$ over $\theta'$.*

The bound on the expected average loss in Theorem 3.6 boils down to $\Lambda^*/c + (L/(c\beta))||\theta_1 - \theta^*||^2/n$ by taking $\rho = 1/(2L)$. Notably, under the condition on the sampling probability in Theorem 3.6, the convergence rate is of order $O(1/n)$. A similar bound is known to hold for SGD with adaptive stochastic Polyak step size for the finite-sum problem, as seen in Theorem 3.4 of Loizou et al. [2021]. A difference is that Theorem 3.6 allows for sampling of the points.

**Loss and Uncertainty-based Sampling for Linear Binary Classifiers**   We consider linear binary classifiers, focusing particularly on logistic regression and the binary cross-entropy training loss function. The following corollaries of Theorem 3.6 hold for sampling proportional to absolute error loss and an uncertainty-based sampling probability function, respectively.

**Corollary 3.7.** *For sampling proportional to absolute error loss, $\pi(u) = \omega(1 - \sigma(u))$, with $\beta/(4(1-c)L') \leq \omega \leq 1$ and $\rho = 1/(2L)$, the bound on the expected loss in Theorem 3.6 holds.*

**Corollary 3.8.** *For the uncertainty-based sampling according to*

$$\pi(u) = \frac{\beta}{2(1-c)}\min\left\{\rho R^2\frac{1}{H(a) + (1-a)|u|}, 1\right\}$$

*where $a \in (0, 1/2]$ and $H(a) = a\log(1/a) + (1-a)\log(1/(1-a))$, the bound on the expected loss in Theorem 3.6 holds.*

**Other Cases**   For a constant sampling probability function with a value of at least $\kappa/(2(1-c))$, condition (6) holds when $\kappa \leq 2(1-c)$. When $\pi(x, y, \theta) = \zeta(x, y, \theta)^\eta$, where $\eta \geq 0$ and $\rho\beta \in (0, 1]$, condition (6) holds under $\beta^{1-\eta} \leq 2(1-c)(1/(2L))^\eta$, as shown in Appendix A.14. Condition (6) also holds for an uncertainty-based sampling in multi-class classification, as shown in Appendix A.15.

## 4   Numerical Results

In this section we evaluate our AWS algorithm, defined in Section 3.2. In particular, we focus on an instance of AWS with stochastic Polyak's expected step size for logistic regression and the loss-based sampling proportional to absolute error loss, which we refer to as *Adaptive-Weight Sampling - Polyak Absloss (AWS-PA)*. By, Corollary 3.7, AWS-PA converges according to Theorem 3.6. Here we demonstrate convergence on real-world datasets and compare with other algorithms.

The implementation of AWS-PA algorithm along with all the other code run the experimental setup that is described in this section is available at `https://www.github.com/facebookresearch/AdaptiveWeightSampling`.

We use a modified version of the *mushroom* binary classification dataset [Chang and Lin, 2011] that was used by Loizou et al. [2021] for evaluation of their algorithm. This modification uses RBF kernel features, resulting in a linearly separable dataset for a linear classifier like logistic regression. Furthermore, we include five datasets that we selected at random from the 44 real-world datasets that were used in Yang and Loog [2018], a benchmark study of active learning for logistic regression: *MNIST 3 vs 5* LeCun et al. [1998], *parkinsons* Little et al. [2007], *splice* Noordewier et al. [1990], *tictactoe* Aha [1991], and *credit* Quinlan [1987]. While these datasets are not necessarily linearly separable, Yang and Loog [2018] has shown that logistic regression achieves a good quality-of-fit.

In our evaluation, we deliberately confine the training to a single epoch. Throughout this epoch, we sequentially process each data instance, compute the loss for each individual instance, and subsequently update the model's weights. This approach, known as *progressive validation* [Blum et al., 1999], enables us to monitor the evolution of the average loss. The constraint to a single epoch ensures that we calculate losses only for instances that haven't influenced model weights. For each sampling scheme, we conduct a hyper-parameter sweep to minimize the average progressive loss and apply a procedure to ensure that all algorithms sample comparable numbers of instances.

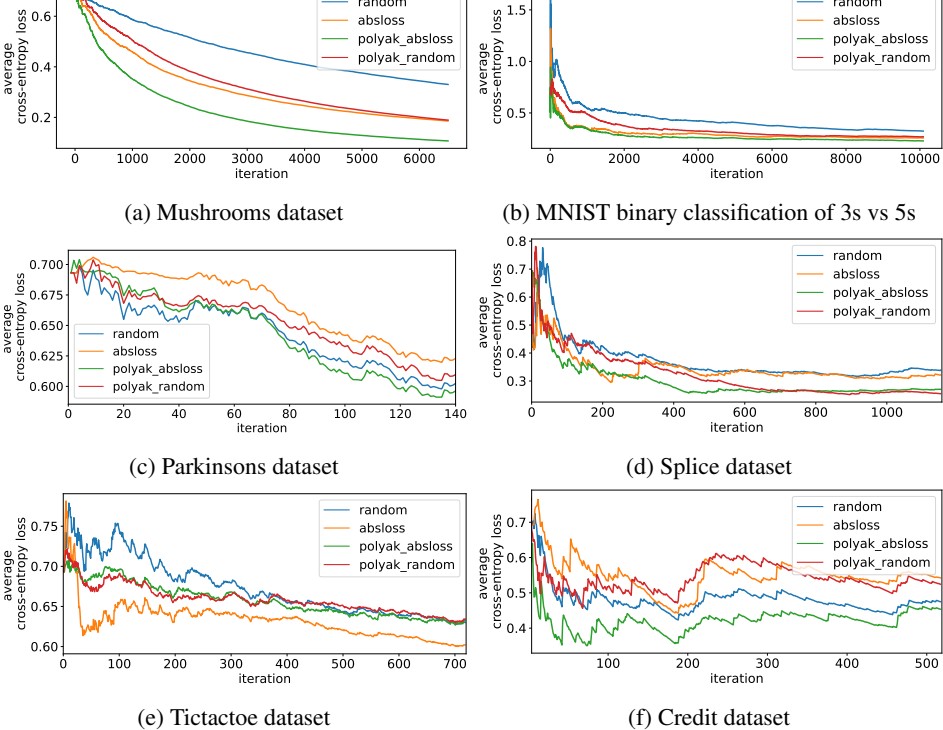

Figure 1: Convergence in terms of average cross-entropy progressive loss of random sampling, loss-based sampling based on the absolute error loss, and our proposed algorithm (loss-based sampling with stochastic Polyak's step size). Our proposed algorithm outperforms the baselines in most cases.

In Appendix B.1 we include further details on this procedure, the hyper-parameter tuning, and other aspects of the experimental setup.

Figure 1 demonstrates that AWS-PA leads to faster convergence than the traditional loss-based sampling with a constant step size (akin to Yoo and Kweon [2019]). It also shows that the traditional loss-based sampling approach converges more rapidly than random sampling on five of the six datasets. These results are obtained under a hyper-parameter tuning such that different algorithms have comparable data sampling rates. We provide additional experimental results that demonstrate the efficiency of AWS-PA in Appendix B.2.

In active learning applications, the true loss of a point cannot be computed before the corresponding label is obtained. Hence, in practice we do not know the true loss at the moment of making the sampling decision. Therefore, we assess the effect of using a *loss estimator*, instead of using the true loss values. We use a Random Forest regressor to estimate absolute error loss based on the same set of features as the target model and the target's model prediction as an extra feature. We retrain this estimator on every sampling step using the labeled points observed so far.

Figure 2 demonstrates that AWS-PA with the estimated absolute error losses performs similarly on all datasets to AWS-PA with the true absolute error losses. Moreover, for a majority of the datasets, the two variants of AWS-PA achieve similar data sampling rates; this is shown Appendix B.3 along with further discussion.

## 5 Conclusion

We have provided convergence rate guarantees for loss and uncertainty-based active learning algorithms under various assumptions. Furthermore, we introduced the novel *Adaptive-Weight Sampling (AWS)* algorithm that combines sampling with an adaptive size, conforming to stochastic Polyak's step size in expectation, and demonstrated its convergence rate guarantee, contingent on a condition related to the sampling probability function.

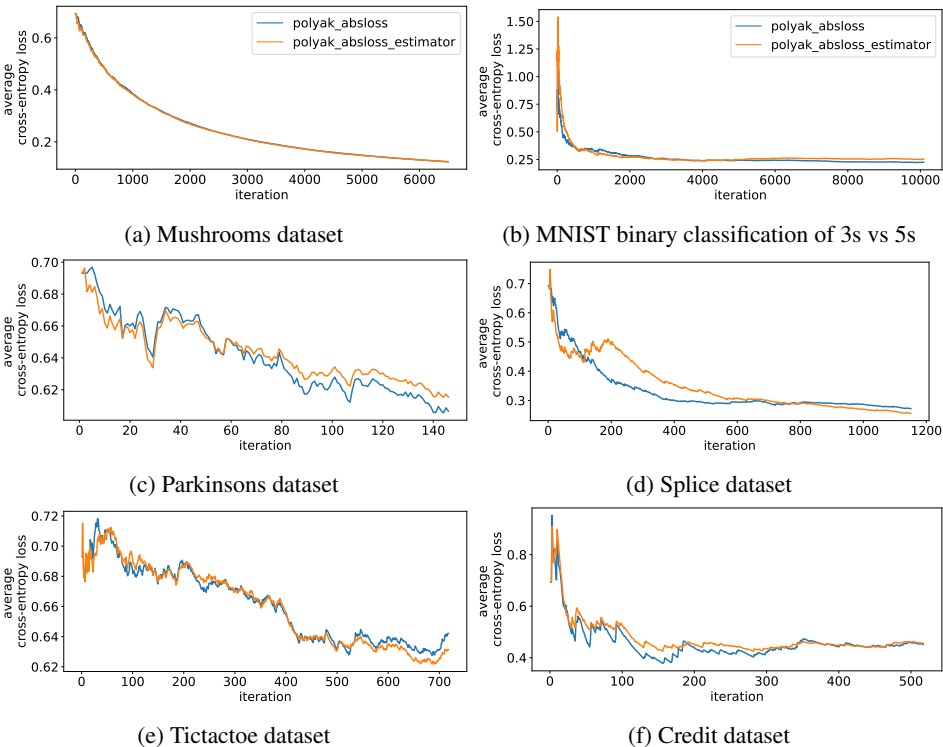

Figure 2: Active learning sampling based on an estimator of the absolute error loss performs on par with the sampling based on the ground truth value of absolute error loss.

For future research, it would be interesting to establish tight convergence rates for the training loss function and the sampling cost, especially comparing policies using sampling with a constant probability with those using adaptive loss-based sampling probabilities. It would be interesting to explore adaptive-weight sampling algorithms with adaptive sizes different than those studied in this paper. Additionally, exploring a theoretical study on the impact of bias and noise in the loss estimator, used for evaluating the sampling probability function, on the convergence properties of algorithms could open up a valuable avenue for investigation.

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

# A    Appendix: Proofs, Discussion, and Additional Results

## A.1    Limitations & Discussion

It remains an open problem to investigate the tightness of convergence rate bounds for constant-weight sampling under the assumptions outlined in Theorem 3.3.

The convergence rate results presented in Theorems 3.3 and 3.6 pertain to smooth convex training loss functions. Future research may explore weaker assumptions regarding the training loss function.

Regarding loss-based sampling strategies, our theoretical analysis assumes an unbiased and noiseless loss estimator. Extending this to account for estimation bias and noise would be a valuable avenue for further investigation.

Our numerical results demonstrate the effectiveness of our proposed algorithm and the robustness of our theoretical findings to loss estimation bias and noise across different datasets, utilizing the logistic regression model as a binary classifier. Future work could explore the application of other classification models, such as multi-layer neural networks.

## A.2    A Set of Convergence Rate Conditions

We present and prove two convergence rate lemmas for algorithm (1): the first providing sufficient conditions for a certain convergence rate and the second restricted to linear classifiers and linearly separable datasets.

The first lemma relies on the following condition involving the loss function $\ell$ used by algorithm (1) and the loss function $\tilde{\ell}$ used for evaluating the performance of the algorithm.

**Assumption A.1.** *There exist constants $\alpha, \beta > 0$ such that for all $(x,y) \in \mathcal{X} \times \mathcal{Y}$ and $\theta \in \Theta$,*

$$\pi(x,y,\theta)||\nabla_\theta \ell(x,y,\theta)||^2 \leq \alpha\tilde{\ell}(x,y,\theta) \tag{7}$$

*and*

$$\pi(x,y,\theta)\nabla_\theta \ell(x,y,\theta)^\top(\theta - \theta^*) \geq \beta\tilde{\ell}(x,y,\theta). \tag{8}$$

**Lemma A.2.** *Under Assumption A.1, for any $\theta_1$ and $\{\theta_t\}_{t>1}$ according to algorithm (1) with $\gamma = \beta/\alpha$,*

$$\mathbb{E}\left[\sum_{t=1}^n \tilde{\ell}(x_t, y_t, \theta_t)\right] \leq ||\theta_1 - \theta^*||^2 \frac{\alpha}{\beta^2}.$$

*Moreover, if for every $(x,y) \in \mathcal{X} \times \mathcal{Y}$, $\tilde{\ell}(x,y,\theta)$ is a convex function in $\theta$, then we have*

$$\mathbb{E}[\tilde{\ell}(x,y,\bar{\theta}_n)] \leq ||\theta_1 - \theta^*||^2 \frac{\alpha}{\beta^2}\frac{1}{n}$$

*where $\bar{\theta}_n = (1/n)\sum_{t=1}^n \theta_t$.*

*Proof.* For any $\theta^* \in \Theta$ and $t \geq 1$, we have

$$||\theta_{t+1} - \theta^*||^2 = ||\theta_t - \theta^*||^2 - 2z_t\nabla_\theta \ell(x_t, y_t, \theta_t)^\top(\theta_t - \theta^*) + z_t^2||\nabla_\theta \ell(x_t, y_t, \theta_t)||^2.$$

Taking expectation in both sides of the equation, conditional on $x_t$, $y_t$ and $\theta_t$, we have

$$\begin{aligned}\mathbb{E}[||\theta_{t+1} - \theta^*||^2 \mid x_t, y_t, \theta_t] = {}& ||\theta_t - \theta^*||^2 - 2\gamma\pi(x_t, y_t, \theta_t)\nabla_\theta \ell(x_t, y_t, \theta_t)^\top(\theta_t - \theta^*) \\ & + \gamma^2\pi(x_t, y_t, \theta_t)||\nabla_\theta \ell(x_t, y_t, \theta_t)||^2.\end{aligned}$$

Under Assumption A.1, we have

$$\mathbb{E}[||\theta_{t+1} - \theta^*||^2 \mid x_t, y_t, \theta_t] \leq ||\theta_t - \theta^*||^2 - 2\gamma\beta\tilde{\ell}(x_t, y_t, \theta_t) + \gamma^2\alpha\tilde{\ell}(x_t, y_t, \theta_t).$$

Hence, it holds

$$(2\gamma\beta - \gamma^2\alpha)\mathbb{E}[\tilde{\ell}(x_t, y_t, \theta_t)] \leq \mathbb{E}[||\theta_t - \theta^*||^2] - \mathbb{E}[||\theta_{t+1} - \theta^*||^2].$$

Summing over $t$, we have

$$(2\gamma\beta - \gamma^2\alpha)\sum_{t=1}^{n}\mathbb{E}[\tilde{\ell}(x_t, y_t, \theta_t)] \le ||\theta_1 - \theta^*||^2.$$

By taking $\gamma = \beta/\alpha$, we have

$$\sum_{t=1}^{n}\mathbb{E}[\tilde{\ell}(x_t, y_t, \theta_t)] \le ||\theta_1 - \theta^*||^2 \frac{\alpha}{\beta^2}.$$

The second statement of the lemma follows from the last above inequality and Jensen's inequality. $\square$

For the case of linear classifiers and linearly separable datasets, it can be readily checked that Lemma A.2 implies the following lemma.

**Lemma A.3.** *Assume that there exist constants $\alpha, \beta > 0$ such that for all $u \in \mathbb{R}$,*

$$\pi(u)\ell'(u)^2 R^2 \le \alpha\tilde{\ell}(u) \tag{9}$$

*and*

$$\pi(u)(-\ell'(u))(\rho^* - u) \ge \beta\tilde{\ell}(u). \tag{10}$$

*Then, for any $\theta_1$ such that $||\theta_1 - \theta^*|| \le S$ and $\{\theta_t\}_{t>1}$ according to algorithm (1) with $\gamma = \beta/\alpha$,*

$$\mathbb{E}\left[\sum_{t=1}^{n}\tilde{\ell}(y_t x_t^\top \theta_t)\right] \le S^2 \frac{\alpha}{\beta^2}.$$

*Moreover, if $\tilde{\ell}$ is a convex function, then $\mathbb{E}\left[\tilde{\ell}(yx^\top\bar{\theta}_n)\right] \le S^2\alpha/(\beta^2 n)$ where $\bar{\theta}_n = (1/n)\sum_{t=1}^{n}\theta_t$.*

### A.2.1 Discussion

We discuss some implications of conditions (9) and (10). Some of this discussion will help us identify the types of loss functions to which the conditions cannot be applied.

First, we note that $\tilde{\ell}(u) > 0$ implies $\pi(u) > 0$. Hence, equivalently, $\pi(u) = 0$ implies $\tilde{\ell}(u) = 0$.

Second, we note that under conditions (9) and (10) it is necessary that for all $u \in \mathbb{R}$, either

$$-\ell'(u) \le \frac{\alpha}{\beta R^2}\max\{\rho^* - u, 0\} \text{ or } \pi(u) = 0.$$

Hence, whenever $\tilde{\ell}(u) > 0$ (and thus $\pi(u) > 0$), then $-\ell'(u) \le \frac{\alpha}{\beta R^2}\max\{\rho^* - u, 0\}$. The latter condition means that $\ell'(u) = 0$ whenever $u \ge \rho^*$ and otherwise the derivative of $\ell$ at $u$ is bounded such that $\ell'(u) \ge -(\alpha/(\beta R^2))(\rho^* - u)$. In other words, function $\ell$ must not decrease too fast on $(-\infty, \rho^*]$.

Third, assume $\ell$ and $\tilde{\ell}$ are such that $\ell(u) = 0$ and $\tilde{\ell}(u) = 0$ for all $u \ge 1$ and $\tilde{\ell}(u) > 0$ for all $u < 1$. Then, for every $u \le 1$

$$\int_u^1 (-\ell(v))dv \le \frac{\alpha}{\beta R^2}\int_u^1 (\rho^* - v)dv$$

which by integrating is equivalent to

$$\ell(u) \le \frac{\alpha}{\beta R^2}\left((\rho^* - 1)(1 - u) + \frac{1}{2}(1 - u)^2\right).$$

This shows that $\ell$ must be upper bounded by a linear combination of hinge and squared hinge loss function.

Forth, assume that $\pi$ is decreasing in $u$, then for every fixed $u_0 \in \mathbb{R}$,

$$\tilde{\ell}(u) \ge \frac{\pi(u_0)R^2}{\alpha}(-\ell'(u))^2 \text{ for every } u \le u_0.$$

If $\tilde{\ell} = \ell$, then

$$\ell(u) \leq \left( \sqrt{\ell(u_0)} + \frac{1}{2R} \sqrt{\frac{\alpha}{\pi(u_0)}} (u_0 - u) \right)^2 \quad \text{for all } u \leq u_0.$$

Fifth, and last, assume that $\pi$ is an even function and that there exists $c > 0$ and $u_0 \leq \rho^*$ such that $\tilde{\ell}(u) \geq c$ for every $u \leq u_0$. Then,

$$\pi(u) = \Omega \left( \frac{1}{|u|^2} \right)$$

which limits the rate at which $\pi(u)$ is allowed to decrease with $|u|$. To see, this, from conditions (9) and (10), for every $u \leq \rho^*$,

$$\tilde{\ell}(u) \leq \frac{\alpha}{\beta^2 R^2} \pi(u)(\rho^* - u)^2.$$

Hence, $\pi(u) \geq (c\beta^2 R^2/\alpha)/(\rho^* - u))$ for every $u \leq u_0$. The lower bound is tight in case when $\ell$ is squared hinge loss function and $\tilde{\ell}(u) = c$ for every $u \leq u_0 < 1$. In this case, from conditions (9) and (10), for every $u \leq u_0$,

$$c\beta \frac{1}{(1 - u)(\rho^* - u)} \leq \pi(u) \leq \frac{c\alpha}{R^2} \frac{1}{(1 - u)^2}$$

which implies $\pi(u) = \Theta(1/|u|^2)$.

## A.3  Proof of Theorem 3.1

We show that conditions of the theorem imply conditions (9) and (10) to hold, for $\tilde{\ell} = \ell$, which in turn imply conditions (7) and (8) and hence we can apply the convergence result of Lemma A.3.

We first consider condition (7). For squared hinge loss function $\ell$ and $\tilde{\ell} = \ell$, clearly condition (7) holds for every $u \geq 1$ as in this case both side of the inequality are equal to zero. For every $u \leq 1$,

$$\frac{\ell(u)}{\ell'(u)^2} = \frac{1}{2}.$$

Since by assumption $\pi(u) \leq \beta/2$ for all $u$, condition $\alpha/\beta \geq R^2$ implies condition (7).

We next consider condition (8). Again, clearly, condition holds for every $u \geq 1$ as in this case both sides of the inequality are equal to zero. For $u \leq 1$, we can write (8) as follows

$$\begin{aligned}
\pi(u) \geq \beta \frac{\ell(u)}{(-\ell'(u))(\rho^* - u)} &= \frac{\beta}{2} \frac{1 - u}{\rho^* - u} \\
&= \frac{\beta}{2} \left( 1 - \frac{1}{1 + (1/(\rho^* - 1))(1 - u)} \right) \\
&= \frac{\beta}{2} \left( 1 - \frac{1}{1 + (\sqrt{2}/(\rho^* - 1))\sqrt{\ell(u)}} \right).
\end{aligned}$$

This condition is implied by $\pi(u) \geq \pi^*(\ell(u))$ where

$$\pi^*(\ell) = \frac{\beta}{2} \left( 1 - \frac{1}{1 + \mu\sqrt{\ell}} \right)$$

with $\mu \geq \sqrt{2}/(\rho^* - 1)$.

The result of the theorem follows from Lemma A.3 with $\alpha/\beta = R^2$, $0 < \beta \leq 2$ and $\pi(u) \geq \pi^*(\ell(u))$ for all $u \leq 1$.

For the expected number of samples we proceed as follows. First by concavity and monotonicity of function $\pi^*$ and the expected loss bound, we have

$$\mathbb{E}\left[ \sum_{t=1}^{n} \pi^*(\ell(x_t, y_t, \theta_t)) \right] \leq \pi^*\left( \mathbb{E}\left[ \frac{1}{n} \sum_{t=1}^{n} \ell(x_t, y_t, \theta_t) \right] \right) n \leq \pi^*\left( \frac{||\theta_1 - \theta^*||^2 R^2}{\beta n} \right) n.$$

Then, combined with the fact $\pi^*(v) \le \frac{\beta\mu}{2}\sqrt{v}$ for all $v \ge 0$, it follows

$$\mathbb{E}\left[\sum_{t=1}^n \pi^*(\ell(x_t, y_t, \theta_t))\right] \le ||\theta_1 - \theta^*|| \frac{\mu\sqrt{\beta}}{2}\sqrt{n}.$$

Since $\pi^*(v) \le \beta/2$ for all $v$, it obviously holds $\mathbb{E}\left[\sum_{t=1}^n \pi^*(\ell(x_t, y_t, \theta_t))\right] \le (\beta/2)n$, which completes the proof of the theorem.

## A.4 Linear Classifiers: Zero-one Loss and Absolute Error Loss-based Sampling

Here we consider training loss functions satisfying:

**Assumption A.4.** *Function $\ell$ is continuously differentiable on $(-\infty, 0]$, convex, and $\ell'(0) \le -c_1$ and $\lim_{u \to -\infty} \ell'(u) \ge -c_2$, for some constants $c_1, c_2 > 0$.*

We consider sampling proportional to either zero-one loss or absolute error loss. The zero-one loss is defined as $\ell_{01}(u) := \mathbb{1}_{\{u \le 0\}} = \mathbb{1}_{\{y \ne \text{sgn}(x^\top \theta)\}}$. The absolute error loss is defined as $\ell_{\text{abs}}(u) := 2\left(\mathbb{1}_{\{u < 0\}} + \frac{1}{2}\mathbb{1}_{\{u=0\}}\right) = |y - \text{sgn}(x^\top \theta)|$. Sampling proportional to zero-one loss is defined as $\pi(u) = \omega\ell_{01}(u)$ for some constant $\omega \in (0, 1]$, while sampling proportional to absolute error loss is defined as $\pi(u) = \omega\ell_{\text{abs}}(u)$ for some constant $\omega \in (0, 1/2]$.

**Theorem A.5.** *Assume that the loss function $\ell$ satisfies Assumption A.4 and $\rho^* > 0$. Then, under sampling proportional to zero-one loss, for any initial value $\theta_1$ such that $||\theta_1 - \theta^*|| \le S$ and $\{\theta_t\}_{t>1}$ according to algorithm (1) with $\gamma = c_1\rho^*/(c_2^2 R^2)$,*

$$\mathbb{E}\left[\sum_{t=1}^n \ell_{01}(y_t x_t^\top \theta_t)\right] \le \frac{c_2^2 R^2 S^2}{c_1^2 \omega} \frac{1}{\rho^{*2}}. \tag{11}$$

*Furthermore, under sampling proportional to absolute error loss and $\gamma = 2c_1\rho^*/(c_2^2 R^2)$, the bound in (11) holds but with an additional factor of 2.*

Proof is provided in Appendix A.4.1.

The bound in (11) is a well-known bound on the number of mistakes made by the perceptron algorithm, of the order $O(1/\rho^{*2})$. It can be readily observed that under sampling proportional to zero-one loss, the expected number of sampled points is bounded by $(c_2/c_1)^2 R^2 S^2/\rho^{*2}$, which also holds for sampling proportional to absolute error loss but with an additional factor of 4.

### A.4.1 Proof of Theorem A.5

We first consider the case when sampling is proportional to zero-one loss. We show that conditions of the theorem imply conditions (9) and (10) to hold, for $\tilde{\ell} = \ell_{01}$ and $\pi = \omega\ell_{01}$, which in turn imply conditions (7) and (8) and hence we can apply the convergence result of Lemma A.3.

Conditions (9) and (10) are equivalent to:

$$\omega\ell'(u)^2 R^2 \le \alpha, \text{ for every } u \le 0$$

and

$$\omega(-\ell'(u))(\rho^* - u) \ge \beta \text{ for every } u \le 0.$$

Since $\ell$ is a convex function $\ell'(u)^2$ is decreasing in $u$ and $(-\ell'(u))(\rho^* - u)$ is decreasing in $u$. Therefore, conditions are equivalent to

$$\omega(\lim_{u \to -\infty}(-\ell'(u)))^2 R^2 \le \alpha$$

and

$$\omega(-\ell'(0))\rho^* \ge \beta.$$

These conditions hold true by taking $\alpha = \omega c_2^2 R^2$ and $\beta = \omega c_1\rho^*$.

We next consider the case when sampling is proportional to absolute error loss. We show that conditions of the theorem imply conditions (9) and (10) to hold, for $\tilde{\ell} = \ell_{01}$ and $\pi = \omega\ell_{\text{abs}}$, which in turn imply conditions (7) and (8) and hence we can apply the convergence result of Lemma A.3.

Conditions (9) and (10) correspond to

$$2\omega(-\ell'(u))^2 R^2 \le \alpha \text{ for every } u < 0 \text{ and } \omega(-\ell'(0))^2 R^2 \le \alpha$$

and

$$2\omega(-\ell'(u))(\rho^* - u) \ge \beta \text{ for every } u < 0 \text{ and } \omega(-\ell'(0))\rho^* \ge \beta.$$

Again, since $(-\ell'(u))^2$ is decreasing in $u$ and $(-\ell'(u))(\rho^* - u)$ is decreasing in $u$, it follows that the conditions are equivalent to

$$2\omega(\lim_{u \to -\infty}(-\ell'(u)))^2 R^2 \le \alpha$$

and

$$\omega(-\ell'(0))\rho^* \ge \beta.$$

Hence, conditions (9) and (10) hold by taking $\alpha = 2\omega c_2^2 R^2$ and $\beta = \omega c_1 \rho^*$.

## A.5 Linear Classifiers: Generalized Smooth Hinge Loss Function

Here we consider the training loss function corresponding to the generalized smooth hinge loss function [Rennie, 2005], defined for $a \ge 1$ as follows:

$$\ell(u) = \begin{cases} \frac{a}{a+1} - u & \text{if } u \le 0 \\ \frac{a}{a+1} - u + \frac{1}{a+1}u^{a+1} & \text{if } 0 \le u \le 1 \\ 0 & \text{otherwise.} \end{cases} \tag{12}$$

This is a continuously differentiable function that converges to the value of the hinge loss function, $\max\{1 - u, 0\}$, as $a$ goes to infinity. The family of loss functions parameterized by $a$ accommodates the smooth hinge loss function with $a = 1$, introduced by Rennie and Srebro [2005].

**Theorem A.6.** *Assume that $\rho^* > 1$, the loss function is the generalized smooth hinge loss function, and the sampling probability is according to the function $\pi^*$, which, for $\beta \in (0,1]$ and $\rho \in (1, \rho^*]$, is defined as*

$$\pi^*(u) = \begin{cases} \beta \frac{\frac{a}{a+1} - u}{\rho - u} & \text{if } u \le 0 \\ \beta \frac{\frac{a}{a+1}\frac{1}{1-u^a}(1-u) - \frac{1}{a+1}u}{\rho - u} & \text{if } 0 \le u \le 1 \\ 0 & \text{if } u \ge 1. \end{cases}$$

*Then, for any initial value $\theta_1$ such that $\|\theta_1 - \theta^*\| \le S$ and $\{\theta_t\}_{t>1}$ according to algorithm (1) with $\gamma = 1/(c_{a,\rho}R^2)$, where $c_{a,\rho} = a/(a(\rho - 1) + 1)$, we have*

$$\mathbb{E}\left[\ell(yx^\top \bar\theta_n)\right] \le \mathbb{E}\left[\frac{1}{n}\sum_{t=1}^{n}\ell(y_t x_t^\top \theta_t)\right] \le c_{a,\rho}\frac{R^2 S^2}{\beta}\frac{1}{n}.$$

*Furthermore, the same bound holds for every $\pi$ such that $\pi^*(u) \le \pi(u) \le \beta$ for all $u \in \mathbb{R}$, with $c_{a,\rho} = 2a$.*

Proof is provided in Appendix A.5.1.

Note that $\pi^*$ is increasing in $a$ and is upper-bounded by $\pi^{**}(u) = \beta(1 - u)/(\rho - 1)$, which may be regarded as the limit for the hinge loss function. See Figure 3 for an illustration.

We remark that the expected loss bound in Theorem A.6 is the same as for the squared hinge loss function in Theorem 3.1, except for an additional factor $c_{a,\rho} = a/(a(\rho - 1) + 1)$. This factor is increasing in $a$ but always lies in the interval $[1/\rho, 1/(\rho-1)]$, where the boundary values are achieved for $a = 1$ and $a \to \infty$, respectively.

**Theorem A.7.** *Assuming $\rho^* > 1$, the loss function is the generalized smooth hinge loss, and the sampling probability function $\pi^*$ satisfies the assumptions in Theorem A.6, with $n > ((a + 1)/a)c_{a,\rho}R^2 S^2/\beta$, the expected number of sampled points is bounded as*

$$\mathbb{E}\left[\sum_{t=1}^{n}\pi^*(y_t x_t^\top \theta_t)\right] \le \kappa \max\left\{\frac{\sqrt{\beta c_{a,\rho}}RS}{(\rho - 1)\sqrt{a}}\sqrt{n}, \frac{c_{a,\rho}R^2 S^2}{\rho - 1}\right\},$$

*where $\kappa$ is some positive constant.*

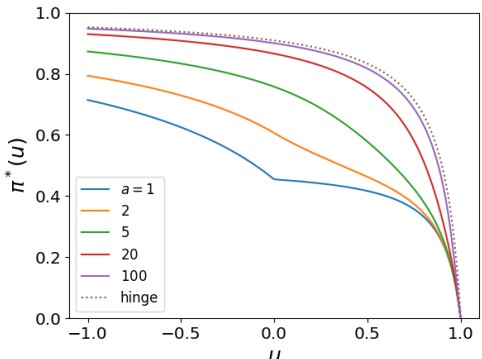

Figure 3: Sampling probability function for the family of generalized smooth hinge loss functions.

Proof is provided in Appendix A.5.2.

For any fixed number of iterations $n$ satisfying the condition of the theorem, the expected number of sampled points is bounded by a constant for sufficiently large value of parameter $a$. The bound in Theorem A.7 depends on how the loss function $\ell(u)$ varies with $u$. When $a$ is large, $\ell(u)$ is approximately $1 - u$ (hinge loss), otherwise, it is approximately $\frac{a}{2}(1 - u)^2$ for $0 \le u \le 1$ (squared hinge loss).

### A.5.1 Proof of Theorem A.6

Assume that $\pi^*$ is such that for given $\rho \in (1, \rho^*]$, $\pi^*(u)(-\ell'(u))(\rho - u) = \beta\ell(u)$ for all $u \in \mathbb{R}$. Then, $\pi^*$ satisfies equation (10) for all $u \in \mathbb{R}$. Note that

$$\pi^*(u) = \beta\frac{\ell(u)}{(-\ell'(u))(\rho - u)} = \begin{cases} \beta\frac{\frac{a}{a+1} - u}{\rho - u} & \text{if } u \le 0 \\ \beta\left(\frac{a}{a+1}\frac{1}{1-u^a}(1 - u) - \frac{1}{a+1}u\right)\frac{1}{\rho - u} & \text{if } 0 \le u \le 1 \\ 0 & \text{if } u \ge 1. \end{cases}$$

By condition (9), we must have

$$\frac{-\ell'(u)}{\rho - u} \le \frac{\alpha}{\beta R^2}.$$

Note that

$$\ell'(u) = \begin{cases} -1 & \text{if } u \le 0 \\ -(1 - u^a) & \text{if } 0 \le u \le 1 \\ 0 & \text{otherwise}. \end{cases}$$

Hence, for every $u \le 0$, it must hold

$$\frac{1}{\rho - u} \le \frac{\alpha}{\beta R^2}$$

which is equivalent to $\alpha/\beta \le R^2/\rho$. For every $0 \le u \le 1$, it must hold

$$f_a(u) := \frac{1 - u^a}{\rho - u} \le \frac{\alpha}{\beta R^2}.$$

Thus, it must hold $\alpha/\beta \ge c_{a,\rho}R^2$ where $c_{a,\rho} = \sup_{u \in [0,1]} f_a(u)$.

Function $f_a$ has boundary values $f_a(0) = 1/\rho$ and $f_a(1) = 0$. Furthermore, note

$$f'_a(u) = \frac{1 + (a - 1)u^a - \rho a u^{a-1}}{(\rho - u)^2}.$$

Note $f_a(0) = 1/\rho^2 > 0$ and $f'_a(1) = -(\rho - 1)a < 0$. Let $u_*$ be such that $f'_a(u_*) = 0$, which holds if, and only if, $g_a(u_*) := (a - 1)u_*{}^a - \rho a u_*{}^{a-1} + 1 = 0$.

Note that

$$c_{a,\rho} = \sup_{u \in [0,1]} f_a(u) = f_a(u_*) = a u_*^{a-1}.$$

For $a = 1$, $f_1$ is decreasing on $[0,1]$ hence $c_{1,\rho} = f_1(0) = 1/\rho$. For $a = 2$, $u_*$ is a solution of a quadratic equation, and it can be readily shown that $c_{2,\rho} = 2\rho(1 - \sqrt{\rho^2 - 1})$. For every $a \geq 1$, we have $c_{a,\rho} \leq a/(1 + a(\rho - 1))$. This obviously holds with equality for $a = 1$, hence it suffices to show that the inequality holds for $a > 1$.

Consider the case $a > 1$. Note that $g_a(u_*) = 0$ is equivalent to

$$a u_*^{a-1} = \frac{1}{\rho - \frac{a-1}{a} u_*}. \tag{13}$$

Combined with the fact $u_* \in [0,1]$, it immediately follows

$$a u_*^{a-1} \leq \frac{a}{a(\rho - 1) + 1}.$$

Furthermore, note that $\lim_{a \to \infty} c_{a,\rho} = 1/(\rho - 1)$. To see this, consider (13). Note that $u_*$ goes to 1 as $a$ goes to infinity. This can be shown by contradiction as follows. Assume that there exists a constant $c \in [0,1)$ and $a_0$ such that $u_* \leq c$ for all $a \geq a_0$. Then, from (13), $a c^{a-1} \geq 1/\rho^*$. The left-hand side in the last inequality goes to 0 as $a$ goes to infinity while the right-hand side is a constant greater than zero, which yields a contradiction. From (13), it follows that $a u_*^{a-1}$ goes to $1/(\rho - 1)$ as $a$ goes to infinity.

We prove the second statement of the theorem as follows. It suffices to show that condition (9) holds true as condition (10) clearly holds for every $\pi$ such that $\pi(u) \geq \pi^*(u)$ for every $u \in \mathbb{R}$. For condition (9) to hold, it is sufficient that

$$f(u) := \frac{\ell(u)}{\ell'(u)^2} \geq \frac{\beta R^2}{\alpha}, \quad \text{for all } u \leq 1.$$

Note that

$$f(u) = \begin{cases} \frac{a}{a+1} - u & \text{if } u \leq 0 \\ \frac{1}{a+1} \frac{u^{a+1} - (a+1)u + a}{(1 - u^a)^2} & \text{if } 0 \leq u \leq 1. \end{cases}$$

Function $f$ is a decreasing function. This is obviously true for $u \leq 0$. For $0 \leq u \leq 1$, we show this as follows. Note that

$$f'(u) = \frac{2a u^{2a} - (2a - 1)(a + 1)u^a + 2a^2 u^{a-1} - (a + 1)}{(a + 1)(1 - u^a)^3}.$$

Hence, $f'(u) \leq 0$ is equivalent to

$$2a u^{a-1}(u^{a+1} + a) \leq (a + 1)(1 - (2a - 1)u^a).$$

In the last inequality, the left-hand side is increasing in $u$ and the right-hand side is decreasing in $u$. Hence the inequality holds for every $u \in [0,1]$ is equivalent to the inequality holding for $u = 1$. For $u = 1$, the inequality holds with equality.

Note that

$$\begin{aligned} f(1) &= \frac{0}{0} \\ &= \frac{1}{a+1} \frac{(a+1)u^a - (a+1)}{-2(1 - u^a)au^{a-1}}\Big|_{u=1} = \frac{0}{0} \\ &= \frac{1}{a+1} \frac{(a+1)au^{a-1}}{2a^2 u^{2(a-1)} - 2(1 - u^a)a(a-1)u^{a-2}}\Big|_{u=1} = \frac{1}{a+1} \frac{(a+1)a}{2a^2} \\ &= \frac{1}{2a}. \end{aligned}$$

It follows that $\inf_{u \leq 1} f(u) = f(1) = \frac{1}{2a} \leq \alpha/(\beta R^2)$, hence it is suffices that $\alpha/\beta \geq R^2/(2a)$.

### A.5.2 Proof of Theorem A.7

**Lemma A.8.** *For every $a \geq 1$,*

$$\pi^*(u) \leq \pi^{**}(u) = \beta \frac{1-u}{\rho-1} \text{ for every } u \leq 1.$$

*Proof.* For $u \leq 1$, $\pi^*(u) = \beta(a/(a+1) - u)/(\rho - u)$, so obviously, $\pi^*(u) \leq \beta(1-u)/(\rho-u)$. For $0 \leq u \leq 1$, we show that next that $a(1-u)/(1-u^a) - u \leq a(1-u)$, which implies that $\pi^*(u) \leq (a/(a+1))\beta(1-u)/(\rho-u)$. To show the asserted inequality, by straightforward calculus it can be shown that the inequality is equivalent to $(1 - (1-u))^{1-a} \geq 1 - (1-a)(1-u)$ which clearly holds true. □

**Lemma A.9.** *For every $c \in (1, 6/5)$, for every $0 \leq u \leq 1$, if $(a+1)(1-u) \geq c$, then*

$$\ell(u) \geq \left(1 - \frac{1}{c}\right)(1-u)$$

*and, otherwise,*

$$\ell(u) \geq \left(1 - \frac{1}{3}\frac{c}{1 - c/2}\right)\frac{a}{2}(1-u)^2.$$

*Proof.* We first show the first inequality. For $u \leq 1$, it clearly holds $\ell(u) \leq 1 - u$, and

$$\frac{\frac{a}{a+1} - u}{1 - u} = \frac{1 - u - \frac{1}{a+1}}{1 - u} = 1 - \frac{1}{(a+1)(1-u)}$$

thus, for every $c > 1$, $\ell(u) \geq (1 - 1/c)(1-u)$ whenever $(a+1)(1-u) \geq c$.

For $0 \leq u < 1$, it holds $\ell(u) = 1 - u - (1/(a+1))(1 - u^{a+1})$, hence it clearly holds $\ell(u) \leq 1 - u$. Next, note

$$\frac{\ell(u)}{1-u} = 1 \frac{1 - u^{a+1}}{(a+1)(1-u)} \geq 1 - \frac{1}{(a+1)(1-u)}.$$

Hence, again, for every $c > 1$, $\ell(u) \geq (1 - 1/c)(1-u)$ whenever $(a+1)(1-u) \geq c$.

We next show the second inequality. For $0 < u \leq 1$, note that $\ell'(u) = -(1 - u^a)$, $\ell''(u) = au^{a-1}$ and $\ell'''(u) = a(a-1)u^{a-2}$. In particular, $\ell'(1) = 0$, $\ell''(1) = a$ and $\ell'''(1) = a(a-1)$. By limited Taylor development, for some $u_0 \in [u, 1]$,

$$\ell(u) = \frac{a}{2}(1-u)^2 - \frac{a(a-1)}{6}u_0^{a-2}(1-u)^3.$$

From this, it immediately follows that $\ell(u) \leq \frac{a}{2}(1-u)^2$ for every $u \leq 1$. For the case $a \geq 2$, we have

$$\ell(u) \geq \frac{a}{2}(1-u)^2\left(1 - \frac{1}{3}(a-1)(1-u)\right).$$

Hence, for every $c \geq 0$, $\ell(u) \geq \frac{a}{2}(1-u)^2(1 - c/3)$ whenever $(a+1)(1-u) \leq c$. For $1 \leq a \leq 2$, we have

$$\ell(u) \geq \frac{a}{2}(1-u)^2\left(1 - \frac{1}{3}\frac{(a-1)(1-u)}{u^{2-a}}\right) \geq \frac{a}{2}(1-u)^2\left(1 - \frac{1}{3}\frac{(a-1)(1-u)}{u}\right).$$

Under $(a+1)(1-u) \leq c$, with $0 \leq c < 2$, we have

$$\frac{(a-1)(1-u)}{u} \leq \frac{c}{1 - \frac{c}{a+1}} \leq \frac{c}{1 - \frac{c}{2}}.$$

Hence, it follows

$$\ell(u) \geq \frac{a}{2}(1-u)^2\left(1 - \frac{1}{3}\frac{c}{1 - \frac{c}{2}}\right).$$

□

Next, note that for every $x, y, \theta$,

$$\pi^*(yx^\top\theta) \leq \pi^{**}(yx^\top\theta) \leq \frac{\beta}{\rho^*-1}(1-yx^\top\theta) = \frac{\beta}{\rho^*-1}(1-\ell^{-1}(\ell(x,y,\theta)))$$

where $\ell^{-1}$ is the inverse function of $\ell(u)$ for $u < 1$ and $\ell^{-1}(0) = 0$.

Hence, we have

$$
\begin{aligned}
\mathbb{E}\left[\sum_{t=1}^n \pi^*(y_t x_t^\top \theta_t)\right] &\leq \frac{\beta}{\rho-1}\left(1-\ell^{-1}\left(\mathbb{E}\left[\frac{1}{n}\sum_{t=1}^n \ell(x_t, y_t, \theta_t)\right]\right)\right)n \\
&\leq \frac{\beta}{\rho-1}\left(1-\ell^{-1}\left(c_{a,\rho}\frac{||\theta_1-\theta^*||^2 R^2}{\beta}\frac{1}{n}\right)\right)n
\end{aligned}
$$

where the first inequality is by concavity of the function $1-\ell^{-1}(\ell)$ and the second inequality is by Theorem A.6.

To apply Lemma A.9, we need that

$$\ell^{-1}\left(c_{a,\rho}\frac{||\theta_1-\theta^*||^2 R^2}{\beta}\frac{1}{n}\right) > 0$$

which is equivalent to

$$c_{a,\rho}\frac{||\theta_1-\theta^*||^2 R^2}{\beta}\frac{1}{n} < \ell(0) = \frac{a}{a+1}$$

i.e.

$$n > \frac{a+1}{a}c_{a,\rho}\frac{||\theta_1-\theta^*||^2 R^2}{\beta}.$$

By Lemma A.9, we can distinguish two cases when

$$1-\ell^{-1}\left(c_{a,\rho}\frac{||\theta_1-\theta^*||^2 R^2}{\beta}\frac{1}{n}\right) \geq \frac{c}{a+1}$$

or, otherwise, where $c$ is an arbitrary constant in $(1, 6/5)$.

In the first case,

$$1-\ell^{-1}\left(c_{a,\rho}\frac{||\theta_1-\theta^*||^2 R^2}{\beta}\frac{1}{n}\right) \leq c_1 c_{a,\rho}\frac{||\theta_1-\theta^*||^2 R^2}{\beta}\frac{1}{n}$$

where $c_1 = 1/(1-1/c)$ while in the second case

$$1-\ell^{-1}\left(c_{a,\rho}\frac{||\theta_1-\theta^*||^2 R^2}{\beta}\frac{1}{n}\right) \leq c_2\sqrt{\frac{2}{a}}\sqrt{c_{a,\rho}\frac{||\theta_1-\theta^*||^2 R^2}{\beta}\frac{1}{n}}$$

where $c_2 = 1/(1-(1/3)c/(1-c/2))$.

It follows that for some constant $\kappa > 0$,

$$\mathbb{E}\left[\sum_{t=1}^n \pi^*(y_t x_t^\top \theta_t)\right] \leq \kappa \max\left\{\frac{\sqrt{\beta c_{a,\rho}}}{(\rho-1)\sqrt{a}}||\theta_1-\theta^*||R\sqrt{n}, \frac{c_{a,\rho}}{\rho-1}||\theta_1-\theta^*||^2 R^2\right\}.$$

### A.6 Multi-class Classification for Linearly Separable Data

We consider multi-class classification with $k \geq 2$ classes. Let $\mathcal{Y} = \{1, \ldots, k\}$ denote the set of classes. For every $y \in \mathcal{Y}$, let $\theta_y \in \mathbb{R}^d$ and let $\theta = (\theta_1^\top, \ldots, \theta_k^\top)^\top \in \mathbb{R}^{kd}$ be the parameter. For given $x$ and $\theta$, predicted class is an element of $\arg\max_{y\in\mathcal{Y}} x^\top\theta_y$.

The linear separability condition is defined as follows: there exists $\theta^* \in \mathbb{R}^{kd}$ such that for some $\rho^* > 1$, for every $x \in \mathcal{X}$ and $y \in \mathcal{Y}$,

$$x^\top\theta_y^* - \max_{y'\in\mathcal{Y}\backslash\{y\}} x^\top\theta_{y'}^* \geq \rho^*.$$

Let

$$u(x, y, \theta) = x^\top \theta_y - \max_{y' \in \mathcal{Y} \setminus \{y\}} x^\top \theta_{y'}.$$

We consider margin loss functions which are according to a decreasing function of $u(x, y, \theta)$, i.e. $\ell(x, y, \theta) \equiv \ell(u(x, y, \theta))$ and $\tilde{\ell}(x, y, \theta) \equiv \tilde{\ell}(u(x, y, \theta))$. For example, this accomodates hinge loss function for multi-class classification Crammer and Singer [2002].

**Lemma A.10.** *Conditions in Assumption A.1 hold provided that for every $x$, $y$ and $\theta$,*

$$\pi(x, y, \theta)(-\ell'(u(x, y, \theta))^2 2R^2 \leq \alpha \tilde{\ell}(u(x, y, \theta))$$

*and*

$$\pi(x, y, \theta)(-\ell'(u(x, y, \theta)))(\rho^* - u(x, y, \theta)) \geq \beta \tilde{\ell}(u(x, y, \theta)).$$

*Proof.* For $x \in \mathcal{X}$ and $y \in \mathcal{Y}$, let $\phi(x, y) = (\phi_1(x, y)^\top, \ldots, \phi_k(x, y)^\top)^\top$ where $\phi_i(x, y) = x$ if $i = y$ and $\phi_i(x, y)$ is the $d$-dimensional null-vector, otherwise. Note that

$$u(x, y, \theta) = \theta^\top \phi(x, y) - \frac{1}{|\mathcal{Y}^*(x, y, \theta)|} \sum_{y' \in \mathcal{Y}^*(x, y, \theta)} \theta^\top \phi(x, y') \tag{14}$$

where

$$\mathcal{Y}^*(x, y, \theta) = \arg \max_{y' \in \mathcal{Y}^*(x, y, \theta)} \theta_{y'}^\top x.$$

Note that

$$||\nabla_\theta \ell(x, y, \theta)||^2 = \ell'(u(x, y, \theta))^2 ||\nabla_\theta u(x, y, \theta)||^2.$$

From (14),

$$\nabla_\theta u(x, y, \theta) = \phi(x, y) - \frac{1}{|\mathcal{Y}^*(x, y, \theta)|} \sum_{y' \in \mathcal{Y}^*(x, y, \theta)} \phi(x, y').$$

It can be readily shown that

$$||\nabla_\theta u(x, y, \theta)||^2 = \left(1 + \frac{1}{|\mathcal{Y}^*(x, y, \theta)|}\right) ||x||^2 \leq 2||x||^2.$$

Hence, we have

$$||\nabla_\theta \ell(x, y, \theta)||^2 \leq 2(-\ell'(u(x, y, \theta))^2 ||x||^2. \tag{15}$$

Next, note that $\nabla_\theta \ell(x, y, \theta) = \ell'(u(x, y, \theta)) \nabla_\theta u(x, y, \theta)$,

$$\nabla_\theta \ell(x, y, \theta)^\top (\theta - \theta^*) = (-\ell'(u(x, y, \theta))) \nabla_\theta u(x, y, \theta)^\top (\theta^* - \theta).$$

and

$$\nabla_\theta u(x, y, \theta)^\top \theta^* = x^\top \theta_y^* - \max_{y' \in \mathcal{Y} \setminus \{y\}} x^\top \theta_{y'}^* \geq \rho^*.$$

It follows

$$\nabla_\theta \ell(x, y, \theta)^\top (\theta - \theta^*) \geq (-\ell'(u(x, y, \theta)))(\rho^* - u(x, y, \theta)). \tag{16}$$

Using (15) and (16), for conditions (7) and (8) to hold, it suffices that

$$\pi(x, y, \theta)(-\ell'(u(x, y, \theta)))^2 2R^2 \leq \alpha \tilde{\ell}(u(x, y, \theta))$$

and

$$\pi(x, y, \theta)(-\ell'(u(x, y, \theta)))(\rho^* - u(x, y, \theta)) \geq \beta \tilde{\ell}(u(x, y, \theta)).$$

Note that these conditions are equivalent to those for the binary case in (9) and (10) except for an additional factor 2 in the first of the last above inequalities. $\qquad\square$

## A.7 Proof of Lemma 3.2

Function $\Pi$ is a convex function because, by assumption, $\pi$ is an increasing function. By (4 and Jensen's inequality, we have

$$
\begin{aligned}
\mathbb{E}\left[\frac{1}{n}\sum_{t=1}^{n}\tilde{\ell}(\theta_t)\right] &= \mathbb{E}\left[\frac{1}{n}\sum_{t=1}^{n}\Pi(\ell(x_t,\theta_t))\right] \\
&\geq \Pi\left(\mathbb{E}\left[\frac{1}{n}\sum_{t=1}^{n}\ell(x_t,\theta_t)\right]\right) \\
&= \Pi\left(\mathbb{E}\left[\frac{1}{n}\sum_{t=1}^{n}\ell(x_t,y_t,\theta_t)\right]\right) \\
&= \Pi\left(\mathbb{E}\left[\frac{1}{n}\sum_{t=1}^{n}\ell(\theta_t)\right]\right).
\end{aligned}
$$

Therefore, we have

$$
\mathbb{E}\left[\frac{1}{n}\sum_{t=1}^{n}\ell(\theta_t)\right] \leq \Pi^{-1}\left(\mathbb{E}\left[\frac{1}{n}\sum_{t=1}^{n}\tilde{\ell}(\theta_t)\right]\right).
$$

Combined with condition (5), we have

$$
\begin{aligned}
\mathbf{E}\left[\frac{1}{n}\sum_{t=1}^{n}\ell(\theta_t)\right] &\leq \Pi^{-1}\left(\inf_{\theta}\tilde{\ell}(\theta) + \sum_{i=1}^{m}f_i(n)\right) \\
&\leq \inf_{\theta}\Pi^{-1}(\tilde{\ell}(\theta)) + \sum_{i=1}^{m}\Pi^{-1}(f_i(n))
\end{aligned}
$$

where the last inequality holds because $\Pi^{-1}$ is a concave function, and hence, it is a subadditive function.

## A.8 Proof of Theorem 3.3

Under assumptions of the theorem, by Theorem 6.3 Bubeck [2015],

$$
\mathbb{E}\left[\tilde{\ell}(\bar{\theta}_n)\right] \leq \mathbb{E}\left[\frac{1}{n}\sum_{t=1}^{n}\tilde{\ell}(\theta_t)\right] \leq \inf_{\theta}\tilde{\ell}(\theta) + \sqrt{2}S\sigma_\pi\frac{1}{\sqrt{n}} + LS^2\frac{1}{n}.
$$

Combining with Lemma 3.2, we obtain the assertion of the theorem.

## A.9 Proof of Corollary 3.4

**Lemma A.11.** *For $\Pi(x) = x - 1 + e^{-x}$,*

$$
\Pi^{-1}(y) \leq 2\sqrt{y} \text{ for } y \in [0, (3/4)^2].
$$

*Proof.* We consider

$$
\Pi(x) = x - (1 - e^{-x}).
$$

By limited Taylor development,

$$
1 - e^{-x} \leq x = \frac{1}{2}x^2 + \frac{1}{6}x^3.
$$

Hence,

$$
\Pi(x) \geq \frac{1}{2}x^2\left(1 - \frac{1}{3}x\right).
$$

Note that $\Pi(x) \geq cx^2$ for some constant $c > 0$ provided that

$$\frac{1}{2}x^2\left(1 - \frac{1}{3}x\right) \geq cx^2$$

which is equivalent to $x \leq 3(1 - 2c)$. Hence, for any fixed $c \in [0, 1/2)$, we have

$$\Pi(x) \geq cx^2, \text{ for every } x \in [0, 3(1 - 2c)].$$

Now, condition $x \leq 3(1 - 2c)$ is implied by $\sqrt{\Pi(x)/c} \leq 3(1 - 2c)$, i.e. $\Pi(x) \leq 9c(1 - 2c)^2$. Hence,

$$\Pi^{-1}(y) \leq \sqrt{\frac{1}{c}y} \text{ for } y \in [0, 9c(1 - 2c)^2].$$

In particular, by taking $c = 1/4$, we have

$$\Pi^{-1}(y) \leq 2\sqrt{y} \text{ for } y \in [0, (3/4)^2].$$

$\square$

We have the bound in Theorem 3.3. Under $\sqrt{2}S\sigma_\pi/\sqrt{n} \leq (3/4)^2$, we have

$$\Pi^{-1}\left(\sqrt{2}S\sigma_\pi\frac{1}{\sqrt{n}}\right) \leq 2^{5/4}\sqrt{S\sigma_\pi}\frac{1}{\sqrt[4]{n}}.$$

Under $LS^2/n \leq (3/4)^2$, we have

$$\Pi^{-1}\left(LS^2\frac{1}{n}\right) \leq 2\sqrt{L}S\frac{1}{\sqrt{n}}.$$

This completes the proof of the corollary.

### A.10  Proof of Theorem 3.6

To simplify notation, we write $\ell_t(\theta) \equiv \ell(x_t, y_t, \theta)$, $\gamma_t = \gamma(x_t, y_t, \theta_t)$ and $\pi_t = \pi(x_t, y_t, \theta_t)$.

Since $\ell$ is an $L$-smooth function, we have $\ell(x, y, \theta) - \inf_{\theta'} \ell(x, y, \theta') \geq 1/(2L^2)||\nabla_\theta\ell(x, y, \theta)||^2$. Hence, for any $x, y, \theta$ such that $||\nabla_\theta\ell(x, y, \theta)|| > 0$, we have

$$\frac{\ell(x, y, \theta) - \min_{\theta'}\ell(x, y, \theta')}{||\nabla_\theta\ell(x, y, \theta)||^2} \geq \frac{1}{2L}. \tag{17}$$

Combined with the definition of $\zeta$ and the fact $\zeta(x_t, y_t, \theta_t) = \gamma_t\pi_t$, we have

$$\kappa := \beta\min\left\{\frac{1}{2L}, \rho\right\} \leq \gamma_t\pi_t \text{ whenever } ||\nabla_\theta\ell(x, y, \theta)|| > 0. \tag{18}$$

From the definition of $\zeta$ and the fact $\zeta(x_t, y_t, \theta_t) = \gamma_t\pi_t$, we have

$$\gamma_t\pi_t \leq \rho\beta. \tag{19}$$

Next, note

$$\begin{aligned}
&\mathbb{E}[||\theta_{t+1} - \theta^*||^2 \mid x_t, y_t, \theta_t] \\
=\ & ||\theta_t - \theta^*||^2 - 2\mathbb{E}[z_t \mid x_t, y_t, \theta_t]\nabla_\theta\ell_t(\theta_t)^\top(\theta_t - \theta^*) + \mathbb{E}[z_t^2 \mid x_t, y_t, \theta_t]||\nabla_\theta\ell_t(\theta_t)||^2 \\
=\ & ||\theta_t - \theta^*||^2 - 2\gamma_t\pi_t\nabla_\theta\ell_t(\theta_t)^\top(\theta_t - \theta^*) + \gamma_t^2\pi_t||\nabla_\theta\ell_t(\theta_t)||^2 \\
\leq\ & ||\theta_t - \theta^*||^2 - 2\gamma_t\pi_t(\ell_t(\theta_t) - \ell_t(\theta^*)) + \gamma_t^2\pi_t\frac{||\nabla_\theta\ell_t(\theta_t)||^2}{\ell_t(\theta_t) - \ell_t(\theta_t^*)}(\ell_t(\theta_t) - \ell_t(\theta_t^*)) \\
=\ & ||\theta_t - \theta^*||^2 - \gamma_t\pi_t\left(2 - \gamma_t\frac{||\nabla_\theta\ell_t(\theta_t)||^2}{\ell_t(\theta_t) - \ell_t(\theta_t^*)}\right)(\ell_t(\theta_t) - \ell_t(\theta_t^*)) + 2\gamma_t\pi_t(\ell_t(\theta^*) - \ell_t(\theta_t^*)) \\
\leq\ & ||\theta_t - \theta^*||^2 - 2c\kappa(\ell_t(\theta_t) - \ell_t(\theta_t^*)) + 2\rho\beta(\ell_t(\theta^*) - \ell_t(\theta_t^*))
\end{aligned}$$

where the first inequality is by convexity of $\ell$, the second inequality is by condition (6), and (18) and (19). Hence, we have

$$\mathbb{E}[||\theta_{t+1} - \theta^*||^2] \leq \mathbb{E}[||\theta_t - \theta^*||^2] - 2c\kappa_1\mathbb{E}[\ell_t(\theta_t) - \ell_t(\theta_t^*)] + 2\kappa_0\mathbb{E}[\ell_t(\theta^*) - \ell_t(\theta_t^*)].$$

By summing over $t$ from 1 to $n$, we have

$$
\begin{aligned}
\mathbb{E}\left[\frac{1}{n}\sum_{t=1}^{n}(\ell_t(\theta_t) - \ell_t(\theta_t^*))\right] &\leq \frac{\rho\beta}{c\kappa}\mathbb{E}\left[\frac{1}{n}\sum_{t=1}^{n}(\ell_t(\theta^*) - \ell_t(\theta_t^*))\right] + \frac{1}{2c\kappa}||\theta_1 - \theta^*||^2\frac{1}{n} \\
&\leq \frac{\rho\beta}{c\kappa}(\mathbb{E}[\ell(x, y, \theta^*)] - \mathbb{E}[\inf_\theta \ell(x, y, \theta)]) + \frac{1}{2c\kappa}||\theta_1 - \theta^*||^2\frac{1}{n}.
\end{aligned}
$$

## A.11 Proofs of Corollaries 3.7 and 3.8

For linear classifiers,

$$\frac{||\nabla_\theta \ell(x, y, \theta)||^2}{\ell(x, y, \theta)} = h(yx^\top\theta)||x||^2$$

where $h(u) = \ell'(u)^2/\ell(u)$ which plays a pivotal role in condition (6).

For the condition (6) to hold it suffices that

$$\pi(x, y, \theta) \geq \frac{\beta}{2(1 - c)}\min\left\{\rho R^2 h(yx^\top\theta), 1\right\}.$$

Note that under assumption that $\ell(u)$ is an $L'$-smooth function in $u$, $\ell(yx^\top\theta)$ is an $L'||x||^2$-smooth function in $\theta$. Taking $\rho = 1/(2L)$ with $L = L'R^2$, we have $\rho R^2 = 1/(2L')$.

For the binary cross-entropy loss function, we have $h(u) = \sigma'(u)^2/(\sigma(u)^2(-\log(\sigma(u))))$. Specifically, for the logistic regression case

$$h(u) = \frac{1}{(1 + e^u)^2\log(1 + e^{-u})} \tag{20}$$

which is increasing in $u$ for $u \leq 0$ and is decreasing in $u$ otherwise. Note that $h(u) = (1 - e^{-\ell(u)})^2/\ell(u)$.

### A.11.1 Proof of Corollary 3.7

We first note the following lemma, whose proof is provided in Appendix A.12.

**Lemma A.12.** *Function $h$, defined in (20), satisfies*

$$h(u) \leq \frac{1}{1 + e^u} = 1 - \sigma(u) \text{ for all } u \in \mathbb{R}. \tag{21}$$

*Furthermore $h(u) \sim 1 - \sigma(u)$ for large $u$.*

See See Figure 4, left, for a graphical illustration.

By Lemma A.12, condition (6) in Theorem 3.6 is satisfied with $\rho = 1/(2L)$, by sampling proportional to absolute error loss $\pi^*(u) = \omega(1 - \sigma(u))$ with $\beta/(4(1 - c)L') \leq \omega \leq 1$.

### A.11.2 Proof of Corollary 3.8

We have the following lemma, whose proof is provided in A.13.

**Lemma A.13.** *Function $h$, defined in (20), satisfies, for every fixed $a \in (0, 1/2]$,*

$$h(u) \leq \frac{1}{H(a) + (1 - a)|u|} \text{ for all } u \in \mathbb{R} \tag{22}$$

*where $H(a) = a\log(1/a) + (1 - a)\log(1/(1 - a))$. Furthermore, $h(u) \sim 1/|u|$ as $u$ tends to $-\infty$.*

See See Figure 4, right, for a graphical illustration.

By Lemma A.13, it follows that condition (6) in Theorem 3.6 is satisfied by uncertainty sampling according to

$$\pi^*(u) = \frac{\beta}{2(1 - c)}\min\left\{\rho R^2\frac{1}{H(a) + (1 - a)|u|}, 1\right\}.$$

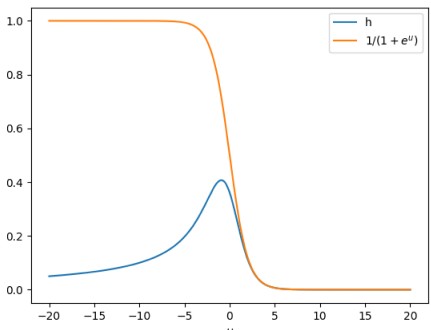 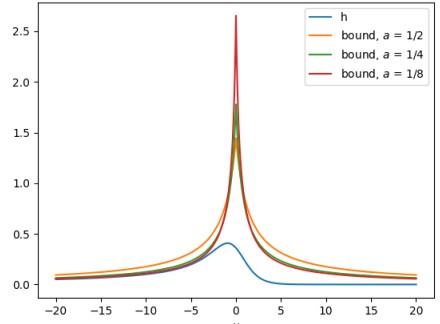

Figure 4: Upper bounds for function $h$ defined in (20): (left) bound of Lemma A.12, (right) bounds of Lemma A.13.

## A.12 Proof of Lemma A.12

We need to prove that for every $u \in \mathbb{R}$,

$$(1 + e^u)^2 \log(1 + e^{-u}) \geq 1 + e^u.$$

By dividing both sides in the last inequality with $(1 + e^u)^2$ and the fact $1/(1 + e^u) = 1 - 1/(1 + e^{-u})$, we note that the last above inequality is equivalent to

$$\log(1 + e^{-u}) \geq 1 - \frac{1}{1 + e^{-u}}.$$

By straightforward calculus, this can be rewritten as

$$\log\left(1 - \left(1 - \frac{1}{1 + e^{-u}}\right)\right) \leq -\left(1 - \frac{1}{1 + e^{-u}}\right).$$

This clearly holds true because $1 - 1/(1 + e^{-u}) \in (0, 1)$ and $\log(1 - z) \leq -z$ for every $z \in (0, 1)$.

It remains only to show that $\lim_{u \to \infty} h(u)/(1 - \sigma(u)) = 1$. This is clearly true as

$$\frac{h(u)}{1 - \sigma(u)} = \frac{1}{(1 + e^u) \log(1 + e^{-u})}$$

which goes to 1 as $u$ goes to infinity.

## A.13 Proof of Lemma A.13

We first consider the case $u \leq 0$. Fix an arbitrary $v \leq 0$. Since $u \mapsto \log(1 + e^u)$ is a convex function it is lower bounded by the tangent passing through $v$, i.e.

$$\log(1 + e^{-u}) \geq \log(1 + e^{-v}) - \frac{1}{1 + e^v}(u - v).$$

Now, let $a$ be such that $1 - a = 1/(1 + e^v)$. Since $v \leq 0$, we have $a \in (0, 1/2]$. It follows that for any fixed $a \in (0, 1/2]$,

$$\log(1 + e^{-u}) \geq H(a) - (1 - a)u.$$

Using this along with the obvious fact $(1 + e^u)^2 \geq 1$, we have that for every $u \leq 0$,

$$h(u) \leq \frac{1}{\log(1 + e^{-u})} \leq \frac{1}{H(a) + (1 - a)|u|}.$$

We next consider the case $u \geq 0$. It suffices to show that for every $u \geq 0$, $h(u) \leq h(-u)$, and hence the upper bound established for the previous case applies. The condition $h(u) \leq h(-u)$ is equivalent to

$$\frac{1}{(1 + e^u)^2 \log(1 + e^{-u})} \leq \frac{1}{(1 + e^{-u})^2 \log(1 + e^u)}.$$

By straightforward calculus, this is equivalent to

$$f(u) := (1 - e^{-2u}) \log(1 + e^u) - u \geq 0.$$

This holds because function (i) $f$ is increasing on $[0, u_0]$ and decreasing on $[u_0, \infty)$, for some $u_0 \geq 0$, (ii) $f(0) = 0$ and (iii) $\lim_{u \to \infty} f(u) = 0$. Properties (ii) and (iii) are easy to check. We only show that property (i) holds true. By straightforward calculus,

$$f'(u) = e^{-2u}(2 \log(1 + e^u) - e^u).$$

It suffices to show that there is a unique $u^* \in \mathbb{R}$ such that $f'(u^*) = 0$. For any such $u^*$ it must hold $2 \log(1 + e^{u^*}) - e^{u^*}$. Let $v = e^{v^*}$. Then, $2 \log(1 + v) = v$, which is equivalent to

$$1 + v = e^{\frac{v}{2}}.$$

Both sides of the last equation are increasing in $v$, and the left-hand side is larger than the right-hand side for $v = 1$. Since the right-hand side is larger than the left-hand side for any large enough $v$, it follows that there is a unique point $v$ at which the sides of the equation are equal. This shows that there is a unique $u^* \geq 0$ such that $f'(u^*) = 0$.

It remains to show that $\lim_{u \to -\infty} h(u)/(1/|u|) = 1$, i.e.

$$\lim_{u \to -\infty} \frac{-u}{(1 + e^u)^2 \log(1 + e^{-u})} = 1$$

which clearly holds true as both $1/(1 + e^u)^2$ and $-u/\log(1 + e^{-u})$ go to 1 as $u$ goes to $-\infty$.

## A.14 Convergence Conditions for $\pi(x, y, \theta) = \zeta(x, y, \theta)^\eta$

It suffices to show that under given conditions, the sampling probability function satisfies condition (6). Using the definition of the sampling probability function, condition (6) can be written as follows

$$\left( \frac{\ell(x, y, \theta) - \inf_{\theta'} \ell(x, y, \theta')}{||\nabla_\theta \ell(x, y, \theta)||^2} \right)^\eta \geq \frac{1}{2(1 - c)} \min \left\{ \beta, \rho\beta \frac{||\nabla_\theta \ell(x, y, \theta)||^2}{\ell(x, y, \theta) - \inf_{\theta'} \ell(x, y, \theta)} \right\}^{1-\eta}. \quad (23)$$

In the inequality (23), by (17), the left-hand side is at least $(1/(2L))^\eta$ and clearly the right-hand side is at most $\beta^{1-\eta}/(2(1 - c))$. Hence, it follows that it suffices that

$$\left( \frac{1}{2L} \right)^\eta \geq \frac{1}{2(1 - c)} \beta^{1-\eta}.$$

## A.15 Uncertainty-based Sampling for Multi-class Classification

We consider multi-class classification according to prediction function

$$p(y \mid x, \theta) = \frac{e^{x^\top \theta_y}}{\sum_{y' \in \mathcal{Y}} e^{x^\top \theta_{y'}}}, \text{ for } y \in \mathcal{Y}.$$

Assume that $\ell$ is the cross-entropy function. Let

$$u(x, y, \theta) = -\log \left( \sum_{y' \in \mathcal{Y} \setminus \{y\}} e^{-(x^\top \theta_y - x^\top \theta_{y'})} \right).$$

It can be shown that

$$\frac{||\nabla_\theta \ell(x, y, \theta)||^2}{\ell(x, y, \theta)} \leq 2||x||^2 h(u(x, y, \theta))$$

where function $h$ is defined in (20). Hence, condition of Theorem 3.6 holds under

$$\pi(u) \geq \frac{\beta}{2(1 - c)} \min \left\{ 2\rho R^2 h(u), 1 \right\}.$$

For given $\theta$ and $x$, let $\theta_{(1)}, \ldots, \theta_{(k)}$ be an ordering of $\theta_1, \ldots, \theta_k$ such that $x^\top \theta_{(1)} \geq \cdots \geq x^\top \theta_{(k)}$. Sampling according to function $\pi^*$ of the gap $g = |x^\top \theta_{(1)} - x^\top \theta_{(k)}|$,

$$\pi^*(g) = \frac{\beta}{2(1-c)} \min\left\{2\rho R^2 h^*(g), 1\right\},$$

where

$$h^*(g) = \frac{1}{H(a) + (1-a)\max\{g - \log(k-1), 0\}},$$

satisfies condition of Theorem 3.6.

We next provide proofs for assertions made above. The loss function is assumed to be the cross-entropy loss function, i.e.

$$\ell(x, y, \theta) = -\log\left(\frac{e^{x^\top \theta_y}}{\sum_{y' \in \mathcal{Y}} e^{x^\top \theta_{y'}}}\right).$$

Note that we can write

$$\ell(x, y, \theta) = -\left(\phi(x, y)^\top \theta - \log\left(\sum_{y' \in \mathcal{Y}\setminus\{y\}} e^{\phi(x, y')^\top \theta}\right)\right).$$

We consider

$$\frac{||\nabla_\theta \ell(x, y, \theta)||^2}{\ell(x, y, \theta)}$$

which is plays a key role in the condition of Theorem 3.6.

It holds

$$\nabla_\theta \ell(x, y, \theta) = -\left(\phi(x, y) - \frac{\sum_{y' \in \mathcal{Y}\setminus\{y\}} e^{\phi(x, y')^\top \theta} \phi(x, y')}{\sum_{y' \in \mathcal{Y}\setminus\{y\}} e^{\phi(x, y')^\top \theta}}\right)$$

and

$$
\begin{aligned}
||\nabla_\theta \ell(x, y, \theta)||^2 &= \left(1 - \frac{e^{\phi(x, y)^\top \theta}}{\sum_{z \in \mathcal{Y}\setminus\{y\}} e^{\phi(x, z)^\top \theta}}\right)^2 ||x||^2 + \sum_{y' \in \mathcal{Y}\setminus\{y\}} \left(\frac{e^{\phi(x, y')^\top \theta}}{\sum_{z \in \mathcal{Y}\setminus\{y\}} e^{\phi(x, z)^\top \theta}}\right)^2 ||x||^2 \\
&= \left(\left(1 - e^{-\ell(x, y, \theta)}\right)^2 + \sum_{y' \in \mathcal{Y}\setminus\{y\}} \left(e^{-\ell(x, y', \theta)}\right)^2\right) ||x||^2.
\end{aligned}
$$

From the last equation, it follows

$$||x||^2 \left(1 - e^{-\ell(x, y, \theta)}\right)^2 \leq ||\nabla_\theta \ell(x, y, \theta)||^2 \leq 2||x||^2 \left(1 - e^{-\ell(x, y, \theta)}\right)^2.$$

Note that $\ell(x, y, \theta) = \log(1 + e^{-u(x, y, \theta)})$ where

$$u(x, y, \theta) = -\log\left(\sum_{y' \in \mathcal{Y}\setminus\{y\}} e^{-(x^\top \theta_y - x^\top \theta_{y'})}\right).$$

It follows

$$||x||^2 h(u(x, y, \theta)) \leq \frac{||\nabla_\theta \ell(x, y, \theta)||^2}{\ell(x, y, \theta)} \leq 2||x||^2 h(u(x, y, \theta))$$

where $h$ is function defined in (20).

The following equation holds

$$u(x, y, \theta) = \theta_y^\top x - \max_{z \in \mathcal{Y}\setminus\{y\}} x^\top \theta_z - \log\left(\sum_{y' \in \mathcal{Y}\setminus\{y\}} e^{-(\max_{z \in \mathcal{Y}\setminus\{y\}} x^\top \theta_z - x^\top \theta_{y'})}\right).$$

Note that

$$
\begin{aligned}
|u(x,y,\theta)| \quad &\geq \quad |x^\top\theta_y - \max_{z\in\mathcal{Y}\setminus\{y\}} x^\top\theta_z| - \log\left(\sum_{y'\in\mathcal{Y}\setminus\{y\}} e^{-(\max_{z\in\mathcal{Y}\setminus\{y\}} x^\top\theta_z - x^\top\theta_{y'})}\right) \\
&\geq \quad |x^\top\theta_{(1)} - x^\top\theta_{(2)}| - \log(k-1).
\end{aligned}
$$

Combining with Lemma A.13, for every $a \in (0, 1/2]$,

$$
h(u(x,y,\theta)) \leq \frac{1}{H(a) + (1-a)|u|} \leq h^*(|x^\top\theta_{(1)} - x^\top\theta_{(2)}|)
$$

where

$$
h^*(g) = \begin{cases} \frac{1}{H(a)} & \text{if } g \leq \log(k-1) \\ \frac{1}{H(a) - (1-a)\log(k-1) + (1-a)g} & \text{if } g > \log(k-1). \end{cases}
$$

# B   Appendix: Additional Material for Numerical Experiments

## B.1   Further Details on Experimental Setup

**Hyperparameter Tuning**   We used the Tree-structured Parzen Estimator (TPE) [Bergstra et al., 2011] algorithm in the `hyperopt` package [Bergstra et al., 2013] to tune the relevant hyperparameters for each method and minimize the average progressive cross entropy loss. For Polyak absloss and Polyak exponent we set the search space of $\eta$ to $[0.01, 1]$ and the search space of $\rho$ to $[0, 1]$. Note that the values of $\eta$ and $\rho$ influence the rate of sampling.

In line with the typical goal of active learning, we aim to learn efficiently and minimize loss under some desired rate of sampling. Therefore, for every configuration of $\eta$ and $\rho$ we use binary search to find the value of $\beta$ that achieves some target empirical sampling rate.

Observe that if we would not control for $\beta$, then our hyperparameter tuning setup would simply find values of $\eta$ and $\rho$ that lead to very high sampling rates, which is not in line with the goal of active learning. In the hyperparameter tuning we set the target empirical sampling rate to 50%.

**Compute Resources**   All experiments were performed on a single machine with 72 CPU cores and 228 GB RAM. It took us around 2,000 seconds to complete a training run for an AWS-PA with an absloss estimator on Mushrooms dataset, our slowest experiment. The training runs for other datasets and algorithms were considerably faster.

## B.2   Further Details on Numerical Experiments with Different Algorithms

In Section 4, we presented numerical results for comparing AWS-PA with other algorithms. These results are shown in Figure 1. Below are some additional details for these experiments.

**Tuning Sampling Rate**   In Figure 1 we compare Polyak absolute loss sampling to absolute loss sampling and random sampling. In this setting we have no control over the sampling rate of absolute loss sampling. Hence, we first run absolute loss sampling to find an empirical sampling rate of 14.9%. We then again use binary search to find the value of $\beta$ to match this sampling rate with Polyak absolute loss sampling. Again, this setup is conservative with respect to the gains of Polyak absolute loss sampling as $\eta$ and $\rho$ were optimized for a sampling rate of 50%.

**Sampling Efficiency of AWS-PA**   In Figure 1 we had demonstrated on various datasets that AWS-PA leads to faster convergence than the traditional loss-based sampling Yoo and Kweon [2019]. Figure 5 presents results as a function of the number of sampled instances, i.e., the number of labeled instances that were selected for training (i.e., cost). This contrasts Figure 1, which showed on the X-axis the total number of iterations. The results confirm that sampling with AWS-PA not only leads to faster convergence than traditional loss-based sampling when expressed in terms of number of iterations, but also when expressed in the number of sampled instances.

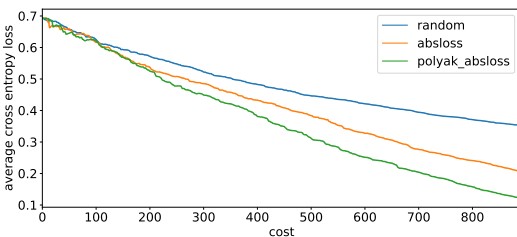

Figure 5: Average cross entropy loss as a function of labeling cost for different sampling methods.

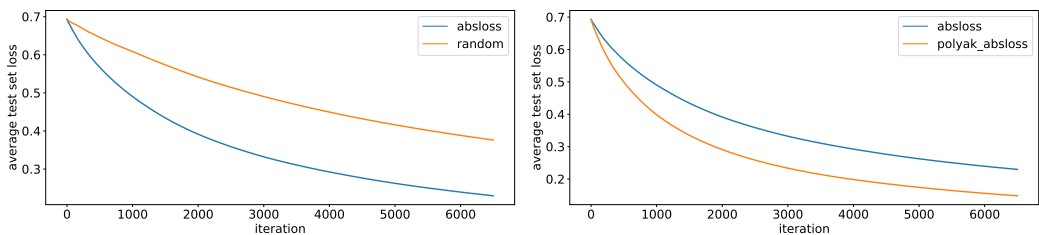

Figure 6: Average cross entropy loss on a hold-out testing set for different sampling methods.

**AWS-PA Results on a Holdout Test Set**    The results in Figure 1 were obtained using a *progressive validation* Blum et al. [1999] procedure where the average loss is measured during an online learning procedure where for each instance the loss is calculated prior to the weight update. Figures 6 and 7 show that our finding that AWS-PA leads to faster convergence than traditional loss-based sampling and than random sampling also holds true on a separate hold out test set.

## B.3    Further Details on the Robustness of AWS-PA to Loss Estimation

In Section 4, we presented numerical results for comparison of the training loss achieved by our AWS-PA algorithm using the ground truth absolute error loss and estimated absolute error loss. These results are shown in Figure 2. Here, we provide more details for the underlying setup of experiments, and the number of sampled points.

**Details on the Absloss Estimator**    For the experiments in Figure 2 we use a separate Random Forest (RF) regressor which estimates absolute error loss based on the same set of features as the target model with an addition of the target's model prediction as an extra feature. The estimator is retrained on every sampling step using the labeled points observed so far. We used the scikit-learn implementation of the RF regressor and manually tuned two hyperparameters for different datasets: (a) number of tree estimators (b) number of "warm-up" steps during which we sample content with a constant probability until we collect enough samples to train an RF estimator. We parallelized training of the RF estimator across all available CPU cores and used default values for all other hyperparameters.

The statistics of the absloss as well as the parameters of the RF estimators for different datasets are summarized in Table 2. From the table, we note that the mean ground truth values of the absloss

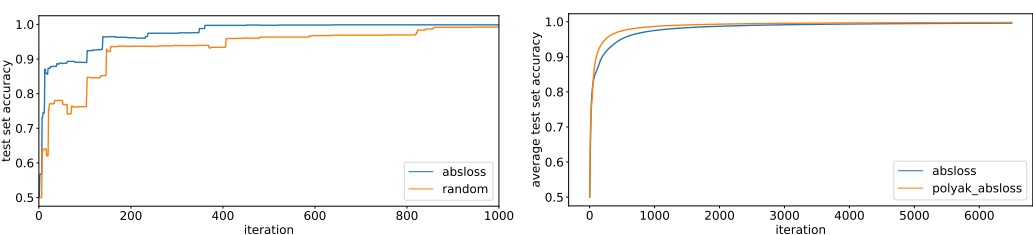

Figure 7: Test accuracy for different sampling methods.

Table 2: Hyperparameters of the absloss estimator and the comparison of the mean of ground truth and the mean of estimated absolute loss values.

| Dataset | Number of trees | Warm-up steps | Mean absloss | Mean estimated absloss |
|---|---|---|---|---|
| Mushrooms | 25 | 1 | 0.100 | 0.104 |
| MNIST 3s vs 5s | 25 | 50 | 0.089 | 0.087 |
| Parkinsons | 100 | 5 | 0.448 | 0.443 |
| Splice | 100 | 25 | 0.163 | 0.160 |
| Tictactoe | 100 | 1 | 0.435 | 0.416 |
| Credit | 100 | 50 | 0.277 | 0.294 |

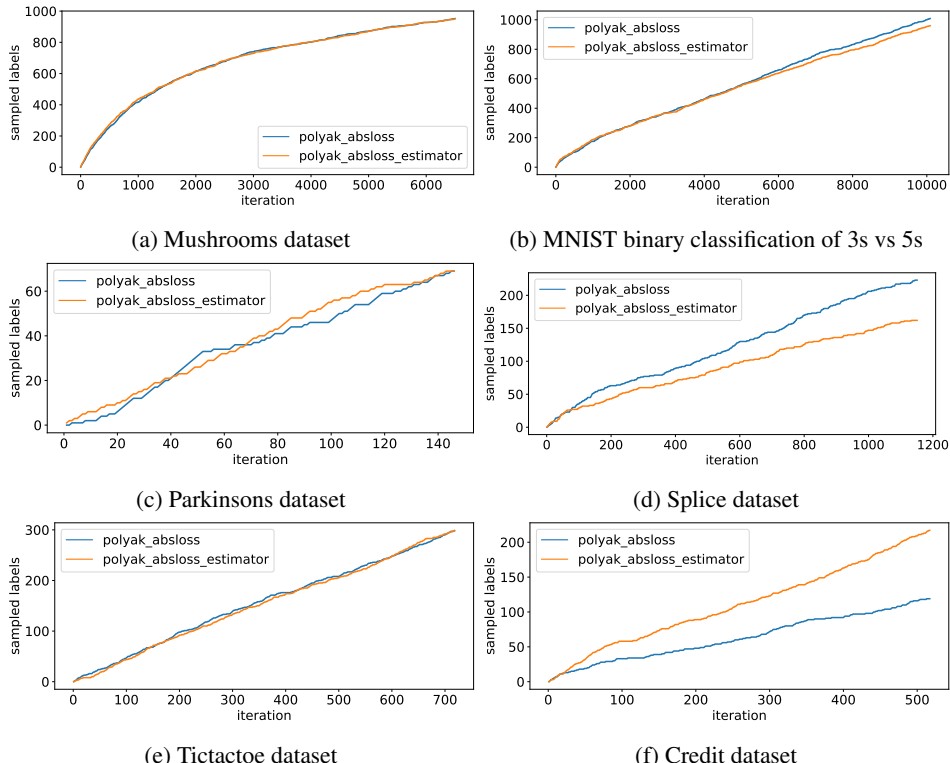

(a) Mushrooms dataset

(b) MNIST binary classification of 3s vs 5s

(c) Parkinsons dataset

(d) Splice dataset

(e) Tictactoe dataset

(f) Credit dataset

Figure 8: Sampling efficiency for sampling based on the ground truth of absolute loss v.s. on estimated absolute loss.

are largely in line with the mean estimated absloss. This suggests that it is possible to train absloss estimator with low bias or is even unbiased.

**Sampling efficiency of the absloss estimator**    In Figure 8 we compare the cost of sampling based on the ground truth absolute loss versus sampling based on the estimated absloss. We note that in 4 out 6 datasets, the sampling cost closely matches that of sampling based on the ground truth absloss. However, in one of the cases (Splice) the sampling cost is lower and in one of the cases (Credit) it is higher than the baseline.

**Alternative absloss estimators**    In Figure 2 we shared results of AWS-PA using a Random Forest regressor to estimate the absolute loss. Figure 9 shows the results the credit dataset of an otherwise identical experimental setup where we have replaced the Random Forest regressor absloss estimator with an MLP neural network.

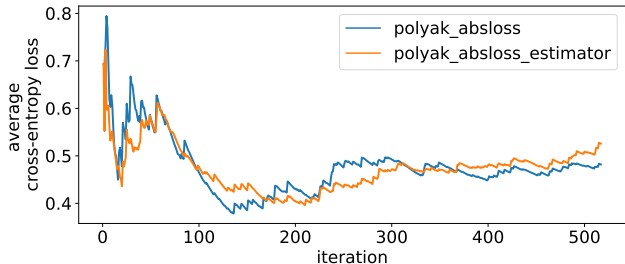

Figure 9: Credit dataset with an MLP neural network loss estimator.

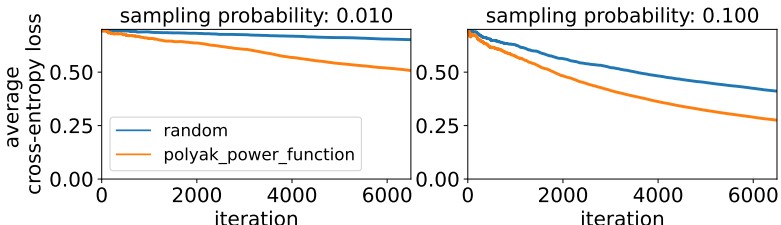

Figure 10: Average cross-entropy progressive loss of Polyak's step size compared to SGD with constant step size, for 1% and 10% sampling from the mushrooms data.

### B.4 Additional experiments

**Experiments with different sampling rates**  Loizou et al. [2021] demonstrated that stochastic gradient descent with a step size corresponding to their stochastic Polyak's step size converges faster than gradient descent. Figure 10 illustrates that these findings extend to scenarios where we selectively sample from the dataset rather than training on the full dataset, and the step size is according to stochastic Polyak's step size only in expectation.

To perform these experiments, similarly to the procedure described in Appendix B.1, we used binary search to find the value of $\beta$ that correspondingly achieves the two target values 1% and 10% with Polyak power function, while using the values of $\eta$ and $\rho$ that were optimised for a sampling rate of 50%. Therefore, our findings of the gains achieved for selective sampling according to stochastic Polyak's step size are likely conservative since $\eta$ and $\rho$ were not optimised for specifically these sampling rates.

**Experiments with synthetic absloss estimator**  We simulate a noisy estimator of the absolute error loss in AWS-PA. We model an unbiased noisy estimator $\hat{\ell}_{\mathrm{abs}}$ of the absolute error loss $\ell_{\mathrm{abs}} \in [0, 1]$ as a random variable following the beta distribution, denoted as $\hat{\ell}_{\mathrm{abs}} \sim \mathrm{Beta}(\alpha, \beta)$, where $\alpha$ and $\beta$ are parameters set to ensure $\mathbb{E}[\hat{\ell}_{\mathrm{abs}}] = \ell_{\mathrm{abs}}$. The variance of the noise can be controlled by the tuning parameter $\alpha$, given by

$$\mathrm{var}[\hat{\ell}_{\mathrm{abs}}] = \frac{\ell_{\mathrm{abs}}(1 - \ell_{\mathrm{abs}})}{\alpha + \ell_{\mathrm{abs}}}.$$

Figure 11 shows that the convergence results are robust against estimation noise of the absolute error loss for a wide range of values for $\alpha \geq 1$.

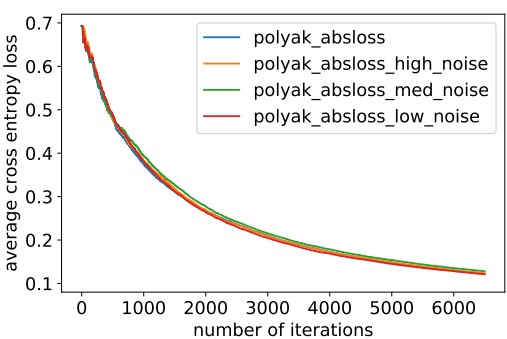

Figure 11: Robustness of the proposed sampling approach with adaptive Polyak's step size for different variance $\text{var}[\hat{\ell}_{\text{abs}}] = \ell_{\text{abs}}^2(1 - \ell_{\text{abs}})/(\alpha + \ell_{\text{abs}})$ noise levels of absolute error loss estimator: (low) $\alpha = 100$, (medium) $\alpha = 2.5$, and (high) $\alpha = 1$.

