# OpenReview forum: "On the Convergence of Loss and Uncertainty-based Active Learning Algorithms"
_NeurIPS.cc/2024/Conference — NeurIPS 2024 poster_

### Official Review · Reviewer_7aAT · 2024-07-02

**Soundness:** 2
**Presentation:** 2
**Contribution:** 2
**Rating:** 5
**Confidence:** 4

**Summary:**

This is a technical paper, whose subject of interest is the convergence of stochastic gradient-based learning algorithms which include a stochastic step size mechanism, whose value is allowed to be influenced by losses or other "uncertainty" related quantities that are computed at training time.

Their main theoretical results can be roughly broken into two categories based on the assumptions placed on the step-size mechanism. The first category is where the step size is a re-scaled Bernoulli random variable, taking values in $\\{0, \\gamma\\}$ in their notation, with $\\gamma$ fixed throughout but the probability of a non-zero step size (i.e., $z\_{t} = \\gamma$) can change depending on the data/parameter at each step in the training routine. They start with an argument centered around a monotonic loss function and linear binary classifiers, but also consider an "equivalent loss" type of strategy (like in Liu an Li '23), again where a convenient monotonicity assumption (here on $\\pi$) preserves convexity and aids in analysis. Their main bounds are in-expectation upper bounds on the loss incurred by the average of iterates generated using this Bernoulli step size.

The second category is similar, but allows the actual step size to be impacted by loss/gradient values in an "adaptive" way, while retaining a certain probability of step size 0. This combination of Bernoulli step sizes with an adaptive step size is what they call "Adaptive-Weight Sampling (AWS)", and they provide conditions to obtain upper bounds on the (empirical) objective function of interest (i.e., the average loss).

Their theoretical results are complemented by empirical analysis, in which they compare uniform random sampling (of points for SGD), "traditional" loss-based sampling, and their AWS approach (their Fig 1). This setup assumes loss access, i.e., this is not active learning. On the other hand, for active learning scenarios, a loss estimator needs to be plugged in; they consider the impact of the quality of such an estimator in their second batch of tests (their Fig 2).

**Strengths:**

Overall, the paper is quite well-written and has a natural logical structure which is easy to follow. The authors have obtained a flexible set of conditions for evaluating SGD procedures with a stochastic loss-dependent step size mechanism, which appear to build quite directly upon the existing literature, which they are good about citing (e.g., Raj and Bach '22, Liu and Li '23).

The paper is a mixture of different efforts, some new convergence theory, a new proposed algorithm (AWS), plus formal/empirical analysis of this algorithm, and I think there is potential for this work to have an audience at a conference like NeurIPS.

**Weaknesses:**

I am not familiar with the related technical literature, so I will not comment on the novelty or theoretical prowess required to obtain the results here.

I would personally highlight two main points I feel need improvement. The first point is that the narrative of this paper feels really bloated. To the best of my reading, all the talk of "active learning" in the title and throughout the paper is totally irrelevant to the entire paper, save for the last paragraph of section 4 plus Figure 2. Yes, there are obvious links between the procedure of interest here and active learning settings, but the core problem setting stands on its own just fine. There is no reason to structure the paper around active learning, it just makes things confusing and downplays the substantive results. I feel like I can say the exact same thing about "uncertainty-based" methods. The only uncertainty-related formulation I can find is Corollary 3.8. Having this is great, but why put uncertainty-based and loss-based methods on the same footing when writing the paper?

The second point is related to technical exposition. For the most part the work seems to be well done, but for a first-time reader, certain parts feel rushed and sloppy. I'll make a list of points I tripped up on in the following section.

**Questions:**

Here are some points that caught my eye while reading the paper. Some are obvious typos, others are poor technical exposition.

- What is the difference with Loizou et al. (2021)? In line 64, the authors say their work is different *"as we consider convergence of SGD under sampling of points."* How is this different? It is unclear.

- Line 140: typo of $\\mathcal{Y} = \\{-,1,1\\}$, should be $\\mathcal{Y} = \\{-1,1\\}$.

- In the key bound of Theorem 3.1 (for example), what is expectation being taken with respect to? On the left-most side of the main inequality, $x$ and $y$ appear. One step to the right, and only $x\_{t}$ and $y\_{t}$ appear. Since there is no generalization analysis, I assumed expectation was with respect to $(x\_{1},y\_{1}),\\ldots,(x\_{n},y\_{n})$ and the randomness in the stochastic algorithm. Is this correct, or are $x$ and $y$ supposed to indicate test data?

- I found the statement on page 5 that *"algorithm (1) is an SGD algorithm with respect to an objective function $\\tilde{\\ell}$ with gradient..."* a bit troubling. Given a random sample of $(x,y)$, indeed $\\pi(x,y,\\theta)\\nabla\_{\\theta}\\ell(x,y,\\theta)$ is an unbiased estimator of $\\nabla\_{\\theta}\\tilde{\ell}(\\theta)$ as defined in (3), but in the case of constant-weight sampling, $z\_{t}$ is *not* equal to $\\pi(x\_{t},y\_{t},\\theta\_{t})$, but rather $\mathbb{E}[z\_{t}] = \\gamma\\pi(x\_{t},y\_{t},\\theta\_{t})$, correct? Perhaps the authors are just glossing over the $\\gamma$, but I think if the authors want to say that $\mathbb{E}[z\_{t}\\nabla\_{\\theta}\ell(x\_{t},y\_{t},\\theta\_{t})]$ equals the right-hand side of (3), it should be done a bit more precisely.

- What is the main difference between the second half of section 3.1 and the work of Liu and Li (2023)? Are they considering the same problem and just looking at one special case of $\\Pi$ and $\\pi$? Is the problem setting different? This is all unclear to me.

- Lemma 3.2: font for expectation is different from the other results ($\\mathbf{E}$ versus $\\mathbb{E}$).

Overall, there is a decent effort here, but I think the paper still needs a fair bit of polish.

**Limitations:**

Not applicable.

---

> ### Author Rebuttal · Authors · 2024-08-05
>
> We thank the reviewer for a well-rounded summary of our results and for recognizing their potential interest to a NeurIPS-like community.
>
> # Weaknesses
> The review focuses on some presentation issues. First, it argues that focusing the paper's exposition around active learning may be confusing and downplays our substantive results (on sampling-based learning). Additionally, it notes that it is unclear why we cover both loss-based and uncertainty-based methods, with the latter discussed only in some parts of the paper. Second, the reviewer noted some technical notation points that require clarification.
>
> The class of algorithms we study (projected SGD with stochastic step size), defined in Section 2, accommodates active learning and data subset selection algorithms by appropriate definition of the stochastic step size.
> The framework of projected SGD with stochastic step size can accommodate different data sampling strategies, including those depending on loss and uncertainty criteria. Our theoretical results in Section 3 apply to active learning learning algorithms under assumption that the algorithm decides whether or not to query the label of a data point based on knowing the exact loss value. Our experimental results evaluate active learning algorithms that decide whether or not to query the label of a data point based on estimated loss value, which showed good conformance to algorithms using exact loss values. Our theoretical results in Section 3 apply to data selection algorithms which can observe the value of the label of each data point, and use this to compute the exact loss value. In our revision of Section 2, we will clarify the connection between active learning algorithms and the projected SGD with stochastic step size--see the concrete revisions we plan to make in Section 2 in our response to reviewer Reviewer ie4P.
>
> As for why covering both loss-based and uncertainty-based methods in our paper, we note that our theoretical framework and results allow us to derive convergence rate bounds for both these methods. Uncertainty-based methods are discussed at several points in our paper and the associated appendix. Following Theorem 3.1 we comment that the proof technique allows to establish the convergence rate for margin uncertainty-based method considered in Raj and Bach [2022]. In Corollary 3.8, we identify an uncertainty-based sampling probability for which the convergence rate bound in Theorem 3.6 holds. Furthermore, in Appendix A.15, we show how this can be extended to a multi-class classification setting.
>
> # Question 1
> The difference is that, unlike Loizou et al [2021], we use *sampling* of points. In contrast, the algorithm by Loizou et al [2021] conducts SGD update with adaptive step size for every input point. Note that this difference is also precisely what makes our work focused on active learning, which focuses on efficient selection/sampling of points. __The comment titled "Proposed clarification for question 1"__ contains the proposed textual clarification for the introduction section. The changes to the problem statement that we propose in response to reviewer ie4P further clarify this.
>
> # Questions 2 & 6
> These are just typos -- we'll fix them.
>
> # Question 3
> In our paper, we consider the streaming computation setting where $(x_1,y_1), \ldots, (x_n,y_n)$ is a sequence of independent and identically distributed labeled data points with distribution $\mathcal{D}$. We will clarify this in the problem statement section (see the revised text in a response to Reviewer ie4P). In Theorem 3.1, $(x,y)$ is an independent sample of a labeled data point from $\mathcal{D}$. Will clarify this by revising the statement of the theorem as __shown in comment titled "Proposed clarification for Question 3"__.
>
> Further details: in Theorem 3.1, we have
> $\mathbb{E}\left[\ell(yx^\top \bar{\theta}\_n)\right] \leq \mathbb{E}\left[\frac{1}{n}\sum\_{t=1}^n \ell(y\_t x\_t^\top \theta\_t)\right].$
>
> Since we consider a streaming algorithm and $(x_1,y_1), \ldots, (x_n, y_n)$ is a sequence of independent labeled data points, for every $t\in \{1,\ldots, n\}$, $(x_t,y_t)$ and $\theta_t$ are independent random variables as $\theta_t$ depends only on $(x_1,y_1),\ldots, (x_{t-1},y_{t-1})$. Hence, it follows that
> \begin{eqnarray*}
> \mathbb{E}\left[\frac{1}{n}\sum_{t=1}^n \ell(y_t x_t^\top \theta_t)\right] &=& \frac{1}{n}\sum_{t=1}^n\mathbb{E}\left[\ell(y_t x_t^\top \theta_t)\right]\\
> &=& \frac{1}{n}\sum_{t=1}^n \mathbb{E}\left[\ell(y x^\top \theta_t)\right]\\
> &=& \mathbb{E}\left[\frac{1}{n}\sum_{t=1}^n \ell(y x^\top \theta_t)\right]\\
> &\geq & \mathbb{E}\left[\ell\left(yx^\top \frac{1}{n}\sum_{t=1}^n \theta_t\right)\right]\\
> &=& \mathbb{E}\left[\ell(yx^\top \bar{\theta}_n)\right]
> \end{eqnarray*}
> where the last inequality follows from Jensen's inequality, as $\ell$ is assumed to be a convex function.
>
>
> # Question 4
> In the context of Equation (3), as noted in the text, $z_t$ is defined by $z_t = \gamma \zeta_t$, where $\zeta_t$ is a Bernoulli random variable with mean $\pi(x_t,y_t,\theta_t)$. This is defined in the text.
>
> # Question 5
> The problem setting is the same. The main difference is that we consider sampling according to a sampling probability function $\pi$, while Liu and Li (2023) considered only some special cases of $\pi$ such as sampling proportional to conditional loss value. We discussed this in the related work Section 1.2.

---

> ### Author Response · Authors · 2024-08-05
> **Proposed clarification for question 1**
>
> We propose to clarify this as follows in the introduction, where the added text is highlighted in bold:
>
>
> > There is a large body of work on convergence of SGD algorithms, e.g. see Bubeck [2015] and Nesterov [2018]. These results are established for SGD algorithms under either constant, diminishing or adaptive step sizes. Recently, Loizou et al. [2021], studied SGD with the stochastic Polyak's step size, depending on the ratio of the loss and the squared gradient of the loss of a point. __Our work proposes an adaptive-window sampling algorithm and provides its convergence analysis, with the algorithm defined as SGD with a sampling of points and an adaptive step size update that conform to the stochastic Polyak's step size in expectation. This is unlike to the adaptive step size SGD algorithm by Loizou et al [2021] which does not use sampling.__

---

> ### Author Response · Authors · 2024-08-05
> **Proposed clarification for Question 3**
>
> Proposed change to Theorem 3.1, changed highlighted in bold.
>
> > Assume that $\rho^* > 1$, the loss function is the squared hinge loss function, and the sampling probability function $\pi$ is such that for all $u \leq 1$, $\pi(u) \leq \beta/2$ and
> >
> > $\pi(u) \geq \pi^*(\ell(u)) := \frac{\beta}{2}\left(1-\frac{1}{1+\mu\sqrt{\ell(u)}}\right)$
> >
> > for some constants $0 < \beta \leq 2$ and $\mu \geq \sqrt{2}/(\rho^*-1)$. Then, for any initial value $\theta_1$ such that $||\theta_1-\theta^*||\leq S$ and $\\{\theta_t\\}_{t>1}$ according to algorithm (1) with $\gamma = 1/R^2$,
> >
> > $\mathbb{E}\left[\ell(yx^\top \bar{\theta}\_n)\right] \leq \mathbb{E}\left[\frac{1}{n}\sum\_{t=1}^n \ell(y\_t x\_t^\top \theta\_t)\right]
>  \leq  \frac{R^2 S^2}{\beta}\frac{1}{n},$
> >
> > __where $(x,y)$ is an independent sample of a labeled data point from $\mathcal{D}$.__
> >
> > Moreover, if the sampling is according to $\pi^*$, then the expected number of sampled points satisfies:
> >
> > $\mathbb{E}\left[\sum\_{t=1}^n \pi^*(\ell(y\_t x\_t^\top \theta\_t))\right] \leq  \min\\{\{\frac{1}{2}R S \mu\sqrt{\beta} \sqrt{n},\frac{1}{2}\beta n\}\\}.$

---

> > ### Comment · Reviewer_7aAT · 2024-08-09
> > **Re: Rebuttal by Authors**
> >
> > I thank the authors for their response. I think that with proper revisions, the paper will be more easily parsed and more accessible to wider audience. I will raise my score.

---

### Official Review · Reviewer_UMCS · 2024-07-12

**Soundness:** 3
**Presentation:** 3
**Contribution:** 3
**Rating:** 6
**Confidence:** 4

**Summary:**

The paper considers the active learning algorithms based on uncertainty and loss functions. The learner queries the label of an unlabeled sample with probability proportional to (some function of) the uncertainty/loss and updates the parameter according to some step size scheme. The authors generalize previous results under the strictly separable binary classification setting and general classification setting with convex loss and smooth equivalent loss. The authors later propose a Polyak-type step size scheme called Adaptive-Weight Sampling and prove its convergence. Numerical experiments verify the efficiency of AWS under both oracle and estimation of loss functions.

**Strengths:**

1. The analysis is solid;
2. The presentation is clear;
3. The Adaptive-Weight Sampling (AWS) algorithm provides a novel perspective for active learning literature.

**Weaknesses:**

1. The generalization of previous results seems not to be very essential;
2. The assumption of access to the exact loss function before querying seems too strong for theoretical analysis.

**Questions:**

1. I wonder if it is possible to relax the requirement of knowing the exact loss functions before querying the label of the sample point.
2. What's the comparison between the theoretical sample complexity of active learning (uncertainty/loss-based sampling) and that of passive learning (uniform sampling) in the considered problem setting?
3. The authors conduct experiments on "uniform sampling" + "constant step size", "loss-based sampling" + "constant step size", and "loss-based sampling" + "Polyak step size" to verify the effectiveness of the approach of loss-based sampling. For completeness, it is necessary to present the performance of using "uniform sampling" + "Polyak step size" in the numerical experiments.
(raised my rating from 5 to 6 after the rebuttal)

**Limitations:**

Yes

---

> ### Author Rebuttal · Authors · 2024-08-05
>
> We thank the reviewer for providing insightful and useful comments.
>
> # Weakness 1
> We would like to clarify the reviewer's comment that we generalize previous results under the strictly separable binary classification setting and the general classification setting with convex loss and smooth equivalent loss. For the linearly separable binary classification setting, we provide new results on the convergence rates for loss-based sampling strategies. Previous work focused on margin of confidence uncertainty-based sampling (Raj and Bach [2022]) which is different. Our results on the convergence rates of constant-weight sampling for "general classification with convex loss" and "smooth equivalent loss" generalize the work by Liu and Li [2023]. Importantly, our convergence rate bounds allow for different sampling probability functions, going beyond specific sampling probability functions such as sampling proportional to a loss value.
>
> # Weakness 2 / Question 1
> For the theoretical convergence rate analysis, it may be possible to relax the requirement of knowing the exact loss function value before querying the label of the sample point. This would involve accounting for the estimation noise of the loss value in the convergence rate analysis. This is an interesting avenue for future research, as indicated in the conclusion section. Note that in our experiments Figure 2, we evaluated algorithms that sample points using a loss value estimator, thus alleviating the need to know the exact loss value before querying the label of the sample point.
>
> # Question 2
> For the question on the comparison of sampling complexities of active learning algorithms and those using uniform sampling of data points, we regard this as an interesting question for theoretical study. Such a study should consider both convergence rate and sampling complexity of algorithms. Please __see the comment below__ for a detailed discussion for the class of projected SGD algorithms using a loss-based sampling.
>
> # Question 3
> We thank the reviewer for this suggestion. We conducted additional experiments to include the case of "uniform sampling" with "Polyak step size." We will include this case in Figure 1 of our revised paper. The new experimental results are provided in the attached PDF file.
>
> From the results, we observe that applying stochastic Polyak's step size with uniform random sampling leads to faster convergence compared to uniform sampling with regular SGD updates. Moreover, sampling according to absolute error loss with stochastic Polyak's step size substantially improves upon uniform sampling with stochastic Polyak's step size.

---

> > ### Comment · Reviewer_UMCS · 2024-08-09
> >
> > Thanks for the detailed rebuttal. The rebuttal addressed some of my concerns, e.g. the response to my Q3.
> >
> > For Weakness 1, I still keep my opinion that the technical novelty is not very essential. I appreciate the part that generalizes Lemma 1 (28) in Raj and Bach's work (https://arxiv.org/pdf/2110.15784) to Assumption A.1, yet the remaining analysis is still the same. Also, for the equivalent loss part, Liu and Li's work has theoretically addressed the necessity of using "a function" ($\Pi$ in your work) of the loss (Section 4.1 in https://arxiv.org/pdf/2307.02719), which makes some of this work's results (e.g. Thm 3.3) predictable.
> >
> > For Weakness 2 and Question 1, maybe your choice of loss-based functions can be justified by comparing yours with more literature (e.g. Towards a statistical theory of data selection under weak supervision, ICLR 2024). I also recommend the authors include more discussions on the cases when the exact loss function is unknown; I'm quite curious if the chosen method (Random Forest) is the best way to estimate the loss.
> >
> > One more question for your response to my Question 2: Does the loss-based sampling have a theoretically better sampling complexity/convergence rate compared to uniform sampling for the general classification setting?
> > Currently, I'm not against accepting this paper (as can be seen from my positive rating); I just want to discuss these questions with the authors.

---

> > > ### Author Response · Authors · 2024-08-12
> > > **Response to Reviewer UMCS's comment on Weakness 1**
> > >
> > > Regarding Weakness 1, in our response, we aimed to emphasize that we provide new results identifying conditions and characterizing the convergence rate and sampling complexity for the linearly separable case. This includes Theorem 3.1 for the squared hinge loss function, as well as several results presented in the appendix, such as Theorem A.5 for sampling proportional to zero-one loss or absolute error loss, and Theorems A.6 and A.7 for a generalized smooth hinge loss function. To the best of our knowledge, these results are novel and provide insights into the performance of loss-based sampling policies.
> > >
> > > As noted by the reviewer, a key technical aspect in the proofs is the convergence rate Lemmas A.2 and A.3, which generalize Lemma 1 in Raj and Bach's work, which is restricted to specific choices of loss functions and uncertainty-based sampling. This generalization was instrumental in establishing our results and may prove useful in future work. We have discussed how our generalization relates to Lemma 1 by Raj and Bach, as seen in lines 196-203. Additionally, note that the proofs of Theorems 3.1, A.5, A.6, and A.7 also require additional technical steps.

---

> > > ### Author Response · Authors · 2024-08-12
> > > **Response to reviewer UMCS's "One more question for your response to my Question 2"**
> > >
> > > Regarding the question of whether loss-based sampling theoretically has a better sampling complexity or convergence rate than uniform sampling in the general classification setting, this is an interesting question that warrants further study. Our experimental results demonstrate instances where sampling based on absolute error loss shows better sampling complexity and convergence rate than uniform sampling.
> > >
> > > The convergence rate result in Theorem 3.3 provides an upper bound on the expected cumulative training loss for loss-based sampling according to a sampling function $\pi$. This bound accommodates uniform sampling as a special case, and in that case, it conforms to the bound that can be obtained through the convergence analysis of projected SGD, as outlined in our response.
> > >
> > > One may compare the convergence rate and sampling complexity bounds for sampling according to a sampling function $\pi$ and uniform sampling, as discussed below.
> > >
> > >
> > > From Theorem 3.3, we have:
> > > $$
> > > \mathbb{E}\left[\frac{1}{n}\sum_{t=1}^n \ell(\theta_t)\right]
> > > \leq \bar{\ell}
> > > $$
> > > where:
> > > $$
> > > \bar{\ell}: = \ell^*_\pi + \Pi^{-1}\left(\frac{\sqrt{2}S\sigma_{\pi}}{\sqrt{n}}\right) + \Pi^{-1}\left(\frac{LS^2}{n}\right)
> > > $$
> > > and
> > > $$
> > > \ell^*_\pi := \inf_{\theta\in \Theta}\Pi^{-1}(\mathbb{E}_x[\Pi(\mathbb{E}_y[\ell(x,y,\theta)\mid x])]).
> > > $$
> > >
> > > For uniform sampling with probability $p\in (0,1]$, we have $\pi(x) = p$, $\Pi(x) = px$ and $\Pi^{-1}(x) = x/p$. Let $\ell^*:=\inf_{\theta \in \Theta} \ell(\theta)$. By the convexity of $\Pi$ (as $\pi$ is an increasing function), note that $\ell^*_\pi\geq \ell^*$ for every $\pi$.
> > >
> > > From Theorem 3.3, for uniform sampling with probability $p \in (0,1]$, it holds that:
> > > $$
> > > \mathbb{E}\left[\frac{1}{n}\sum_{t=1}^n \ell(\theta_t)\right]
> > > \leq  \ell^* + \frac{\sqrt{2}S\sigma}{\sqrt{pn}} + \frac{LS^2}{pn}
> > > $$
> > > where $\sigma$ is such that
> > > $$
> > > \mathbb{E}\_{x,y}[||\nabla\_\theta \ell(x,y,\theta)||^2] - p ||\mathbb{E}\_{x,y}[\nabla_\theta \ell(x,y,\theta)]||^2\leq \sigma^2 \hbox{ for every } \theta \in \Theta.
> > > $$
> > >
> > > For the discussion of sampling complexity, consider the case where $\pi$ is a concave function. By Lemma 3.5, by sampling according to $\pi$, the expected number of samples is upper bounded by $\pi(\bar{\ell})n$. Obviously, for uniform sampling with probability $p$, the expected number of samples is $pn$. Therefore, the sampling complexity for sampling according to the sampling function $\pi$ is lower or equal to that for uniform sampling with probability $p$ if the following condition holds:
> > > $$
> > > p \geq \pi(\bar{\ell}).
> > > $$
> > >
> > > The convergence rate upper bound under sampling according to $\pi$ is smaller than or equal than that under uniform sampling if the following condition holds:
> > > $$
> > > \bar{\ell}\leq \ell^* + \frac{\sqrt{2}S\sigma}{\sqrt{pn}} + \frac{LS^2}{pn}.
> > > $$
> > >
> > > By a straightforward calculus, it can be shown that this condition is equivalent to
> > > $$
> > > \sqrt{pn}\leq S\frac{\sigma + \sqrt{\sigma^2 + 2L(\bar{\ell}-\ell^*)}}{\sqrt{2}(\bar{\ell}-\ell^*)}.
> > > $$
> > > Combining with $p \geq \pi(\bar{\ell})$, it is necessary that
> > > $$
> > > \sqrt{\pi(\bar{\ell})n}\leq S\frac{\sigma + \sqrt{\sigma^2 + 2L(\bar{\ell}-\ell^*)}}{\sqrt{2}(\bar{\ell}-\ell^*)}
> > > $$
> > > which is also sufficient when $p = \pi(\bar{\ell})$. The condition can be further analyzed for different cases including $\ell_\pi^* > \ell^*$ and $\ell_\pi^* = 0$ (and hence $\ell^*=0$).
> > >
> > > In our revision, we will include further discussion on the dependence of the convergence rate and sampling complexity on the sampling function $\pi$. Additionally, we will highlight as an open research problem the need to study the tightness and comparison of convergence rates and sampling complexities for the general classification setting in the conclusion section.

---

> > > > ### Comment · Reviewer_UMCS · 2024-08-12
> > > >
> > > > Thanks for your discussions. The authors discussed abundantly and promised to further portray them in future versions on the "loss-based sampling" principle, which I believe will benefit the active learning community. Therefore, I would increase my rating from 5 to 6.

---

> ### Author Response · Authors · 2024-08-05
> **Detailed response to question 2**
>
> We first discuss convergence rate bounds. For the linearly separable binary classification case, in our paper, we provide conditions under which loss-based sampling policies achieve a convergence rate of $O(1/n)$ for the squared hinge loss function, the same as the perceptron algorithm (Theorem 3.1, and see also Theorems A.5 and A.6 in the appendix). It is well known that standard projected SGD algorithm with a constant step size guarantees a convergence rate of $O(1/\sqrt{n})$ for smooth convex loss functions. This convergence rate can be improved to $O(1/n)$ by using stochastic Polyak's step size, as shown by Loizou et al. [2021]. Our Theorem 3.3 shows that a convergence rate of $O(\Pi^{-1}(1/\sqrt{n}))$ can be achieved by a constant-weight, loss-based sampling with the sampling probability function $\pi$, where $\Pi$ is the primitive function of $\pi$. Our adaptive-weight sampling algorithm guarantees the same convergence rate as the algorithm that samples every point using projected SGD with stochastic Polyak's step size (Loizou et al. [2021]) under conditions given in Theorem 3.6. For our adaptive-weight sampling algorithm, with a constant sampling probability $\pi(x,y,\theta) = p$ for all $x,y,\theta$, the convergence rate bound in Theorem 3.6 holds provided that $p$ satisfies the condition in Equation (6).
>
> We next discuss sampling complexity bounds, i.e., the bounds on the expected number of sampled points by an algorithm. The sampling complexity of an algorithm that samples each point is clearly $n$, where $n$ is the number of SGD updates. This stands in contrast to the sampling complexity of $O(\sqrt{n})$, which is shown to hold for active learning algorithms using certain loss-based policies (Theorem 3.1 and Lemma 3.5). We next consider the case of "uniform sampling," where each data point is sampled with a fixed probability $p$. In this case, the expected number of sampled points is $pn$.
>
> For the linearly separable binary classification case, with uniform sampling with probability $p=\beta/2$, the convergence rate bound in Theorem 3.1 holds and the sampling complexity is $O(\beta n)$. This stands in contrast to the sampling complexity of $O(\sqrt{\beta n})$ under loss-based sampling according to $\pi^*$ defined in Theorem 3.1.
>
> For the general classification setting, we provide the following discussion. For the projected SGD with uniform sampling, we can derive convergence rate bounds by using known results for projected SGD with a constant step size. Consider the projected SGD as in Equation (1) of our paper, with stochastic step size $z_t$ equal to the product of a fixed step size $\gamma > 0$ and $\zeta_t$, where $\zeta_t$ is a sequence of independent Bernoulli random variables with mean $p$. Then, we may regard this as an SGD algorithm with a constant step size $\gamma p$ and the stochastic gradient vector $g_t = (z_t/p) \nabla_\theta \ell(x_t,y_t,\theta_t)$. This stochastic gradient vector is clearly an unbiased estimator of $\nabla\_\theta \ell(x_t,y_t,\theta_t)$ and we have $\mathbb{E}[||g_t - \nabla\_\theta \ell(x\_t,y\_t,\theta\_t))||\_2^2\mid x\_t, y\_t, \theta\_t] = (1/p-1)||\nabla\_\theta \ell(x\_t,y\_t,\theta\_t)||^2$. Thus, we have $\mathbb{E}[||g\_t - \nabla\_\theta \ell(x\_t,y\_t,\theta\_t))||\_2^2\mid x\_t,y\_t,\theta\_t]\leq (1/p-1)\sigma^2$, for $\sigma$ such that $||\nabla \ell(x,y,\theta)||\_2^2\leq \sigma^2$, for every $x,y,\theta$.
> For smooth and convex loss functions, by a well-known convergence result for projected SGD with a constant step size (covered by Theorem 6.3 in Bubeck [2015]), it can be readily shown that for the step size set to $p/(L+(\sigma/S)\sqrt{1/p-1}\sqrt{n/2})$, the following convergence rate bound holds:
> $$
> S\sigma \sqrt{\frac{2(1-p)}{pn}} + \frac{L S^2}{n}
> $$
> where $L$ and $S$ are defined in our paper. Hence, we have the sampling complexity $O(pn)$ and the convergence rate bound $O(1/\sqrt{pn})$. From Theorem 3.3 in our paper, we have the convergence rate bound $O(\Pi^{-1}(\sqrt{2}S\sigma_\pi/\sqrt{n}))$, and with additional assumption that $\pi$ is concave, the sampling complexity $O(\pi(\Pi^{-1}(\sqrt{2}S\sigma_\pi/\sqrt{n}))n)$. For the uniform sampling case, we have $\pi(x) = p$ and $\Pi^{-1}(x) = x/p$. Further noting that $\sigma_\pi \leq \sqrt{p}\sigma$, we have the convergence rate bound of $O(1/\sqrt{np})$ and the sampling complexity of $O(pn)$ holds, both conforming to what we derived above.
>
> In our experimental results (see revised Figure 1 in the attached PDF), we demonstrate that faster convergence can be achieved using loss-based sampling compared to uniform sampling for comparable sampling complexity.

---

> ### Author Response · Authors · 2024-08-12
> **General response to UMCS's comment**
>
> We thank the reviewer for their additional comments, interesting questions for discussion, and useful references. We are glad that the additional experimental results we provided have adequately addressed the concern raised by the reviewer.
>
> We further elaborate on each weakness & question separately in the comments below.

---

> ### Author Response · Authors · 2024-08-12
> **Response to Reviewer UMCS's comment on Weakness 2 and Question 1**
>
> We thank the reviewer for bringing the work by Kolossov, Montanari, and Tandon (2024) to our attention. We will add a discussion of this work in the related work section. Additionally, we will compare the sampling functions studied therein with those considered in our work. The examples of loss-based sampling functions we present are motivated by practical applications, such as the use of absolute error loss in certain industrial contexts.
>
> We will include additional discussion on the case where the exact loss function is unknown. This scenario is addressed in our experimental results, where we used a Random Forest estimator. We believe that the concrete choice for Random Forest isn't of particular importance to our results: the important finding is that there exists regression models that lead to numerical results that show good conformance to those obtained using the exact loss values. Such results could also have been obtained with other alternative choices of the loss estimator (as we show through new experiments that we outline below), and this choice of loss estimator is not of material importance.
>
> It is important to note that proposing a specific loss estimator is beyond the scope of our work. We will provide further discussion on the loss estimator we employed and other estimators we tested.
>
> **Additional experiments with neural network based loss estimator**
>
> Concretely, we have ran additional experiments with a neural network based loss estimator, and like with the RF loss estimator, have obtained numerical results that conform closely to the results that we obtained using the exact loss values.
>
> Per the [NeurIPS 2024 FAQ for authors](https://neurips.cc/Conferences/2024/PaperInformation/NeurIPS-FAQ#:~:text=the%20author%20rebuttal%3F-,No.,all%20linked%20files%20are%20anonymized), we're told not to send links in any part of the response, and unfortunately we do not have the ability to edit the official rebuttal anymore, only post comments, so we can't upload a PDF with the figure with the new results. However, we have sent a message to the AC with an anonymized link to the figure with results with the neural network based loss estimator.

---

### Official Review · Reviewer_ie4P · 2024-07-13

**Soundness:** 3
**Presentation:** 2
**Contribution:** 3
**Rating:** 6
**Confidence:** 2

**Summary:**

This submission studies the convergence guarantees of and bounds on the expected number of samples used when using loss based active learning. They additionally propose a new sampling scheme that combines loss based sampling and a Polyak step size and provide convergence guarantees. Their analysis covers multiple models and loss functions. They proposed methods further evaluated with numerical experiments on multiple datasets.

**Strengths:**

The problem addressed is interesting and has not been addressed in the literature. In order for loss  based sampling strategies to be effectively deployed in the wild this sort of analysis is necessary. The algorithmic contributions are also of interest, and borrow a well known approach to setting step sizes (Polyak step sizes) from the optimization community to define their Adaptive-Weight Sampling scheme. This combination is a novel idea and a starting point to explore other methods of setting step sizes in this active learning context.

**Weaknesses:**

While  the technical contributions of this paper are interesting, the primary weakness is the communication of results. This reviewer has an optimization background rather than an active learning one, but even accounting for this the organization and exposition of results was challenging follow. For example, rather than defining algorithms in a LaTeX algorithm block as is standard in literature, authors simply  refer to modifying the (projected) SGD update. While the general idea of active learning is simple, it is also unclear which variant of active learning the authors are studying. Based on the step size being defined as a Bernoulli variable and the experiments conducting one pass over the dataset, it appears that the authors are studying a “streaming” approach to active learning, where the decision to evaluate the label or not is made upon encountering each datapoint. It appears that the selection operation is to ignore when the Bernoulli sample is zero and evaluate the loss otherwise. This is not made clear in the writing, and an optimization audience may be confused as to why some steps will have step size zero. This understanding could be incorrect, which is likely due to the lack of clarity and motivation of the definitions and algorithms. The authors are encouraged to be more explicit about what problem they are trying to solve, and what the exact definitions of their algorithm is.

**Questions:**

To address the listed lack of clarity, is the above understanding correct?

Aside from the lack of clarity mentioned above, the primary question from an optimization perspective is the choice of the Polyak step size. Given there are many choices for “adaptive” step size methods, it would be interesting to know the motivation behind this choice, and if other methods were considered.

**Limitations:**

The authors have addressed some limitations in their work and potential future directions.

---

> ### Author Rebuttal · Authors · 2024-08-05
>
> We thank the reviewer for appreciating the importance of our work in ensuring the effective deployment of loss-based sampling strategies in practice, and suggesting where we can improve the presentation quality.
>
> # Question 1
> In response to the first question raised by the reviewer: the reviewer's understanding of our problem formulation and algorithm definition is correct. We study convergence rates in the streaming computation setting, where the decision to evaluate the label or not is made upon encountering each data point. This is the same setting as in the convergence analysis of uncertainty-based sampling policies by Raj and Bach [2022]. Our algorithm is defined as the projected SGD in Equation (1) with a stochastic step size ($z_t$ in iteration $t$). Different active learning algorithms are accommodated by appropriately defining the distribution of the stochastic step size $z_t$, allowing for loss- and uncertainty-based sampling. Specifically, we consider constant-weight and adaptive-weight sampling policies, which are explained in Section 2, with further details provided in Section 3 for each algorithm studied. The decision not to evaluate the label of the data point in iteration $t$ implies a stochastic step size of zero ($z_t = 0$). We appreciate the reviewer's comment that a reader with an optimization background may get confused because our setup is similar to but different from the standard projected SGD. The meaning of the stochastic step size in the active learning setting is also different from what is typical in optimization. In our revision, we will add text to emphasize that we study convergence in the streaming computation setting and provide additional explanations for our definitions and algorithms. Specifically, we will make the following modifications in the problem statement section.
>
> ### Revision of Section 2: (added text in bold)
>
> __Consider the setting of streaming algorithms where a machine learning model parameter $\theta_t$ is updated sequentially, upon encountering each data point, with  $(x_1,y_1),\ldots, (x_n,y_n) \in \mathcal{X}\times \mathcal{Y}$ denoting the sequence of data points with the corresponding labels, assumed to be independent and identically distributed with distribution $\mathcal{D}$. Specifically, we consider the class of projected SGD algorithms defined as: given an initial value $\theta_1\in \Theta$,__
>
>
>
> $\theta\_{t+1} = \mathcal{P}\_{\Theta\_0}\left (\theta\_t - z\_t \nabla\_\theta \ell(x\_t, y\_t, \theta\_t)\right), \hbox{for}\hspace{0.1cm}  t \geq 1$  (Equation 1)
>
> where $\ell:\mathcal{X}\times \mathcal{Y}\times \Theta\rightarrow \mathbb{R}$ is a training loss function, $z_t$ is a stochastic step size with mean $\zeta(x_t, y_t, \theta_t)$ for some function $\zeta: \mathcal{X} \times \mathcal{Y} \times \Theta \mapsto \mathbb{R}\_+$, $\Theta\_0 \subseteq \Theta$, and $\mathcal{P}\_{\Theta_0}$ is the projection function, i.e., $\mathcal{P}\_{\Theta_0}(u) = \arg\min\_{v \in \Theta_0} ||u - v||$. Unless specified otherwise, we consider the case $\Theta\_0 = \Theta$, which requires no projection. For binary classification tasks, we assume $\mathcal{Y} = \{-1,1\}$. For every $t>0$, we define $\bar{\theta}\_t = (1/t)\sum\_{s=1}^t \theta\_s$.
>
> __By defining the distribution of the stochastic step  size $z_t$ in Equation 1 appropriately, we can accommodate different active learning and data subset selection algorithms. In the context of active learning algorithms, at each step $t$, the algorithm observes the value of $x_t$ and decides whether or not to observe the value of the label $y_t$ which affects the value of $z_t$. Deciding not to observe the value of the label $y_t$ implies the step size $z_t$ of value zero (not updating the machine learning model).__
>
> For the choice of the stochastic step size, we consider two cases: (a) \emph{Constant-Weight Sampling}: a Bernoulli sampling with a constant step size, and (b) \emph{Adaptive-Weight Sampling}: a sampling that achieves stochastic Polyak's step size in expectation.
>
> For case (a), $z\_t$ is the product of a constant step size $\gamma$ and a Bernoulli random variable with mean $\pi(x\_t, y\_t, \theta\_t)$.
> For case (b), $\zeta(x, y, \theta)$ is the "stochastic" Polyak's step size, and $z\_t$ is equal to $\zeta(x\_t, y\_t, \theta\_t) / \pi(x\_t, y\_t, \theta\_t)$ with probability $\pi(x\_t, y\_t, \theta\_t)$ and is equal to $0$ otherwise. Note that using the notation $\pi(x,y,\theta)$ allows for the case when the sampling probability does not depend on the value of the label $y$.
>
> # Question 2
> With regard to the question on the motivation for studying the adaptive step size according to stochastic Polyak's step size, our primary motivation is its fast convergence rate, as shown by Loizou et al. [2021], both theoretically and experimentally. Specifically, it achieves a convergence rate of $O(1/n)$ in a smooth convex optimization setting, which is an improvement compared to a projected SGD with a constant step size, which achieves a convergence rate of $O(1/\sqrt{n})$. The experimental results in Loizou et al. [2021] demonstrate cases where the adaptive step size according to stochastic Polyak's step size outperforms several benchmarks used for comparison, including the popular Adam optimizer. Our work shows that the aforementioned theoretical convergence rate guarantee of projected SGD with adaptive step size according to stochastic Polyak's step size can be achieved in a data sampling or active learning setting, as shown in Theorem 3.6. Loizou et al. [2021] did not study this setting. As for considering other adaptive step sizes, future work may study their convergence properties when used in conjunction with sampling, as in our adaptive-weight sampling algorithm. We will add a note to propose this as an interesting direction for future research in the conclusion section.

---

> > ### Comment · Reviewer_ie4P · 2024-08-12
> >
> > Thank you to the authors for the clarifications here and in a revision of their paper, along with the insights about their choice of step size. I will maintain my score.

---

### Author Rebuttal · Authors · 2024-08-05

We thank the reviewers for their insightful comments. We appreciate the positive feedback recognizing the problem we study as interesting, novel, and important, as well as the positive assessment of our results, which include both theoretical and experimental analyses. We also value the technical questions and suggestions for improving the clarity of our presentation.

Our study is motivated by the increasing interest in using active learning algorithms to effectively train machine learning models, focusing on the convergence rate of the training loss function and data labeling cost, as well as scalable training of machine learning algorithms by using a subset of training data points. We focus on data selection methods based on data point loss values (loss-based methods)
as those have recently gained attention both in theory (we list several references in the paper) and practice (it is deployed in various industry applications),
while their performance guarantees are not well understood. Our theoretical framework and results also cover more common uncertainty-based methods, where data points are selected based on a notion of prediction uncertainty.

The reviewers expressed different opinions regarding the presentation quality of our paper, with one reviewer finding it clear and others suggesting areas for improvement.
We appreciate the reviewer's feedback and acknowledge that our initial submission lacked clarity, in particular in explaining that we consider a class of streaming algorithms, focusing on projected SGD algorithms with stochastic step size, and how the concept of the stochastic step size relates to active learning and data subset selection problems. A crucial point to understanding this connection is that a zero step size can be seen as not selecting a point for labeling, and hence, the connection to loss-based active learning arises from stochastic step sizes that are specific to each point and depend on its estimated loss. In our review responses, we proposed concrete revisions to clarify these points.

Our theoretical results on convergence rates and sampling complexity of loss-based methods apply to active learning under the assumption that the algorithm has access to the exact loss value of each data point, which is used to decide whether to query the value of a data point's label. In practice, a loss-based active learning algorithm uses a noisy estimate of loss values, which is considered in our experimental results, showing good conformance with our theoretical results. We regard our theoretical results as an important step towards understanding loss-based active learning algorithms. In the context of the data subset selection problem, the algorithm can observe the label of each training data point, and thus exact loss values of data points are available to the algorithm.

A reviewer noted that we missed providing experimental results for one combination of sampling and adaptive step size, namely, uniform sampling with stochastic Polyak's step size. In response, we conducted additional experiments and included the results in our responses (see link to PDF below).

---

### Decision · Program_Chairs · 2024-09-25

**Decision:**

Accept (poster)

**Comment:**

All the reviewers viewed the paper favourably, and it is a creative perspective so I think it would be welcome at the conference.